# Controlled and orthogonal partitioning of large particles into biomolecular condensates

Fleurie M. Kelley [1], Anas Ani[1,2], Emily G. Pinlac[1], Bridget Linders [1], Bruna Favetta[3], Mayur Barai [1], Yuchen Ma[2], Arjun Singh[1], Gregory L. Dignon[1] ✉, Yuwei Gu [2] ✉ & Benjamin S. Schuster [1] ✉

Partitioning of client molecules into biomolecular condensates is critical for regulating the composition and function of condensates. Previous studies suggest that client size limits partitioning. Here, we ask whether large clients, such as macromolecular complexes and nanoparticles, can partition into condensates based on particle-condensate interactions. We seek to discover the fundamental biophysical principles that govern particle inclusion in or exclusion from condensates, using polymer nanoparticles surface-functionalized with biotin or oligonucleotides. Based on our experiments, coarse-grained molecular dynamics simulations, and theory, we conclude that arbitrarily large particles can controllably partition into condensates given sufficiently strong condensate-particle interactions. Remarkably, we also observe that beads with distinct surface chemistries partition orthogonally into immiscible condensates. These findings may provide insights into how various cellular processes are achieved based on partitioning of large clients into biomolecular condensates, and they offer design principles for drug delivery systems that selectively target disease-related condensates.

Condensation of biomolecules enables cells to form compartments without a surrounding membrane. Despite the lack of a delimiting membrane, and despite often exhibiting liquid-like material properties, biomolecular condensates are chemically distinct from the cytoplasm and from one another, giving rise to distinct condensate functions[1–3]. A condensate's biochemical components can be roughly divided into two categories: scaffolds (the biopolymers whose multivalent interactions drive phase separation and condensate formation) or clients (molecules that partition into condensates but which are not required for condensate assembly)[4]. Significant effort has been devoted to understanding which clients partition into condensates, given the importance of this question to condensate biology. Most of these studies have focused on partitioning of small molecules, proteins, and nucleic acids. The view that has emerged is that small molecules may partition into condensates based on chemical compatibility; larger molecules, such as proteins and RNA, encounter additional barriers to partitioning, including that their presence in a condensate reduces the conformational entropy of scaffold biopolymers; but sufficient favorable interactions can permit protein and RNA clients to overcome this entropic penalty and partition into condensates[5–10].

This raises the question: Can even larger particles—on the order of tens or hundreds of nanometers—partition into condensates? Filling in this key knowledge gap is important for understanding how macromolecular assemblies, such as ribosomes, enzyme complexes, and viruses may partition into condensates. The limited existing data are mixed. Studies of dextran partitioning into biomolecular condensates in vitro and in vivo demonstrate a size-exclusion effect, where larger dextrans (especially >70 kDa) tend to be excluded from

[1]Department of Chemical and Biochemical Engineering, Rutgers, The State University of New Jersey, Piscataway, NJ, USA. [2]Department of Chemistry and Chemical Biology, Rutgers, The State University of New Jersey, Piscataway, NJ, USA. [3]Department of Biomedical Engineering, Rutgers, The State University of New Jersey, Piscataway, NJ, USA. ✉e-mail: gregory.dignon@rutgers.edu; yuwei.gu@rutgers.edu; benjamin.schuster@rutgers.edu

condensates[11–13]. On the other hand, recent studies suggest that during HIV-1 infection, the intact virus capsid can transport through the nuclear pore complex (NPC), which is formed by condensation of FG repeat proteins[14,15]. An intermediate case is observed in P granules, where MEG-3 clusters adsorb to the condensate interface[16]. Previous studies have used nanoparticles that partition into condensates as probes of condensate material properties[17–20], and suitable surface functionalization of nanoparticles has been shown to promote particle recruitment into condensates[21]. However, a systematic investigation is required to test whether there is an upper limit to the size of clients that can partition into condensates and to tease apart the biophysical principles that govern whether large clients partition into, are excluded from, or adsorb to the interface of condensates.

The central hypothesis of this study is that large, adhesive particles can partition into condensates via interactions with the condensate scaffold. To test our hypothesis, we engineered a toolbox of nanoparticles of varying sizes and surface chemistries, including particles that resist protein adhesion, particles that bind to scaffold proteins via specific protein-ligand interactions, and particles that interact non-specifically with condensates. We studied the partitioning of these particles into three model in vitro condensates whose phase behavior we and others have previously characterized: the intrinsically disordered RGG domain from LAF-1[17,22], the SARS-CoV-2 nucleocapsid (N) protein[23–25], and a designed polycationic repeat polypeptide (GRGNSPYS)$_{25}$[26].

LAF-1 RGG is a low-complexity sequence rich in Gly, Arg, Asp, Asn, Ser, and Tyr. It has a near-neutral net charge and its phase separation is largely mediated by electrostatic interactions, hydrogen bonds, and cation−π and $sp^2$−π interactions[22]. LAF-1 RGG is representative of intrinsically disordered regions common in biomolecular condensates[26]. N protein is composed of a folded RNA-binding domain and a folded dimerization domain, interspersed among three intrinsically disordered domains[5]. N protein phase separates in association with RNA, and it is representative of RNA-binding proteins abundant in condensates. (GRGNSPYS)$_{25}$ is a highly cationic, artificial intrinsically disordered protein comprising 25 repeats of the octapeptide GRGNSPYS. In the presence of sufficient salt concentration to screen electrostatic repulsion, (GRGNSPYS)$_{25}$ will phase separate, likely due to interactions of the Arg guanidinium group with Tyr and polar residues[26]. (GRGNSPYS)$_{25}$ is interesting because it has a large positive net charge yet does not require a polyanion to phase separate. Thus, these three proteins serve as distinct model systems for studying large-particle partitioning into condensates, and they exemplify three common types of proteins involved in condensate formation: intrinsically disordered domains, RNA-binding proteins, and de novo designed polypeptides.

Here, we conducted experiments, paired with coarse-grained molecular dynamics simulations and theory, to investigate the interplay between particle size and stickiness on partitioning into condensates. Our studies indicate that arbitrarily large particles can partition into condensates based on adhesive interactions between the particle and condensate, but larger particles require greater adhesion strength to do so. Particles that do not interact with condensates are excluded; as the interaction strength increases, the particles localize to the interface of condensates; and as the interaction strength increases yet further, the particles partition into the condensates. Remarkably, large-particle partitioning can be highly tunable and specific, allowing orthogonal partitioning in which two different particle types can target two immiscible condensates. Together, this work addresses the fundamental biophysical question of how size and stickiness determine client partitioning into condensates, which is critical for understanding how condensates regulate their composition in the complex cellular milieu, and it may inform how clients can be engineered to partition into condensates for therapeutic intervention.

## Results

### Partitioning into condensates depends on client size and binding interactions

We first examined the permeability of condensates to various sizes of dextran: 10 kDa, 40 kDa, and 70 kDa, corresponding to hydrodynamic radii of 1.9, 4.8, and 6.5 nm, respectively[27]. We mixed rhodamine-labeled dextrans with the three condensate-forming systems: the LAF-1 RGG domain, the SARS-CoV-2 nucleocapsid (N) protein, and (GRGNSPYS)$_{25}$ (Fig. 1a and Supplementary Fig. 1). As noted above, LAF-1 RGG and (GRGNSPYS)$_{25}$ phase separate in appropriate salt conditions without additional biopolymers[26]. In contrast, the coacervation of N protein is driven by association with RNA. Therefore, unless otherwise noted, all N protein condensates in this study were prepared by adding polyA RNA. We also note that the strong association between N protein and RNA contributes an appreciable elastic rheological component to their condensates, causing them to not be uniformly circular[5]. Based on confocal micrographs, we measured the dextran partition coefficients. We observed that partitioning into condensates is inversely related to dextran size, consistent with previous studies[11,28]. The partition coefficient of 70 kDa dextran was 5-fold lower than that of 10 kDa dextran for all three condensates, and it dropped below 1 for LAF-1 RGG and N protein. These results confirm that size plays an important role in client partitioning into biomolecular condensates, as established in the literature, and suggest that the condensates have a mesh size of very roughly 5 nm[11].

However, hydrophilic and uncharged dextrans are expected to interact only weakly with condensates. We hypothesized that for other clients, interactions with the scaffold protein can drive partitioning into condensates of clients >5 nm. We therefore examined whether larger macromolecular assemblies can partition into condensates. Several studies have hypothesized that actively-translating cellular puncta called translation factories are biomolecular condensates[29–31]. This raises the question of whether ribosomes can partition into biomolecular condensates. To test this, we fluorescently labeled E. coli ribosomes, which are about 21 nm in diameter, and measured their partition coefficient into condensates (Fig. 1b and Supplementary Fig. 2). Ribosomes bound to the periphery of LAF-1 RGG condensates, partitioned heterogeneously into N protein condensates, and partitioned uniformly into (GRGNSPYS)$_{25}$ droplets. These differences can be partially rationalized based on the analysis of the three protein sequences (Supplementary Table 1). The net charge per residue (NCPR) is positive for all three proteins and increases from RGG to N protein to (GRGNSPYS)$_{25}$ (NCPR = 0.017, 0.053, and 0.109, respectively). Meanwhile, the ribosome surface has large areas with negative electrostatic potential[32,33]. These observations suggest that particles significantly larger than the dextrans may partition into condensates, depending on condensate-particle interaction strength.

We next sought to directly test the hypothesis that strong binding interactions between condensate scaffold proteins and clients can drive partitioning, even for large clients. To explore this, we compared partitioning into N protein condensates of antibodies with and without specific affinity for N protein. The approximate molecular weight of IgG is 150 kDa, and its dimensions are about 14.5 × 8.5 × 4.0 nm[34]. Based on our dextran experiments, 70 kDa dextran was mostly excluded from N protein condensates, with a partition coefficient of <0.5. In contrast, despite its larger size, anti-N protein IgG was enriched in N protein condensates, with a partition coefficient of about 5 (Fig. 1c). Isotype control antibody showed only weak partitioning, with a partition coefficient of about 1.5−greater than that of 70 kDa dextran (presumably due to non-specific interactions), but much less than that of anti-N protein IgG. These results further demonstrate that, based on affinity, molecules larger than the tested dextrans can partition into condensates.

Intrigued by the ribosome and antibody partitioning results, we asked whether even larger particles can partition into condensates. We

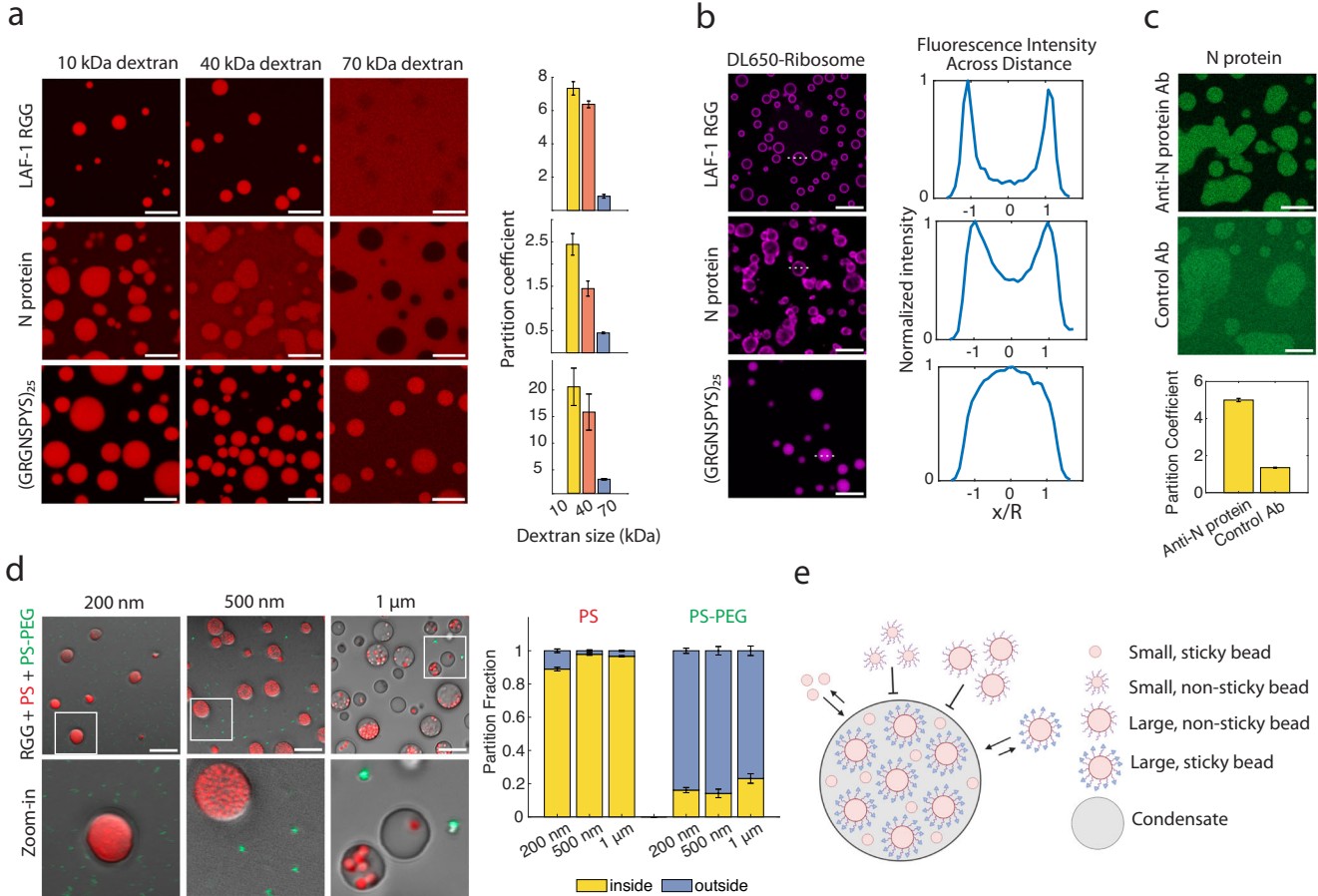

**Fig. 1 | Condensate permeability and partitioning are characterized by a variety of probes. a** 10 kDa, 40 kDa, and 70 kDa rhodamine-labeled dextrans were mixed with LAF-1 RGG; (GRGNSPYS)$_{25}$; or N protein plus polyA RNA. For all figures, buffer was 150 mM NaCl, 20 mM Tris, pH 7.5, unless stated otherwise. (Right) Partition coefficients were quantified for each protein and each dextran size. Data represent mean values and error bars represent standard deviation with $n \geq 11$ images from 2 independent experiments. **b** *E. coli* ribosomes, labeled with DyLight 650 (DL650), were mixed with each protein. Buffer included 1 mM MgCl$_2$. (Right) Line profiles were drawn across $n \geq 30$ condensates in each image, normalized, and averaged. Experiment was repeated independently three times, with similar results. **c** N protein was mixed with anti-N protein IgG or isotype control antibody. (Below) Partition coefficients were quantified for each antibody. Data represent mean values and error bars represent SEM with $n = 12$ images from at least 2 independent

experiments. **d** RGG was mixed with PS (red) and PS-PEG (green) beads of various sizes. PS beads as large as 1 μm partition into RGG condensates. PS-PEG beads are excluded from condensates. (Right) Partition fractions were quantified for each bead type. Any particles inside the condensate or at its interface were counted as inside, and all other particles were counted as outside. Error bars represent SEM with $n = 20$ images from at least 2 independent experiments. **e** Schematic illustrating particle partitioning into condensates. Unmodified PS beads partition into condensates, while PEGylated beads are excluded. Hypothesis: adding sticky moieties, depicted as blue triangles, to PEGylated beads may recruit beads back into condensates. Schematic created in BioRender. Kelley, F. (2025) https://BioRender.com/o27h472. Scale bars represent 10 μm. Source data are provided as a Source Data file.

mixed carboxyl-modified polystyrene (PS) beads of various sizes with LAF-1 RGG protein. Beads even up to 1 μm diameter partitioned into the condensates (Fig. 1d), agreeing with prior microrheology studies[35–37]. (Bead concentration was kept at 0.02 vol% to avoid aggregation and altered condensate morphology observed at higher bead concentration; Supplementary Fig. 3). The PS bead surface is negatively charged due to the carboxylate moieties but retains hydrophobic character from the polystyrene, so multiple non-specific interactions likely contribute to the partitioning of the PS beads into condensates.

Prior studies have demonstrated that coating nanoparticles with a dense brush of low molecular-weight polyethylene glycol (PEG) reduces protein adsorption to the particles[38,39]. Therefore, we hypothesized that PEGylating beads would cause the beads to be excluded from condensates. To test this, we conjugated 5 kDa PEG to the PS beads to form PS-PEG particles (the PS beads are negatively charged, whereas the PS-PEG beads have near-neutral zeta potential; Supplementary Tables 2 and 3). Strikingly, the PS-PEG beads (200 nm, 500 nm, and 1 μm diameter) were essentially completely excluded from the LAF-1 RGG condensates (Fig. 1d). Based on the dichotomy between PS and

PS-PEG bead partitioning, we conclude that sticky beads of arbitrary size can partition into condensates, whereas non-sticky beads are excluded. Our findings clearly indicate that sticky interactions can significantly facilitate the partitioning of particles into condensates. Although the PS beads partition due to non-specific interactions, our results so far suggest that functionalizing particles with biomolecules that have specific interactions with condensates may enable targeted partitioning of clients into these condensates (Fig. 1e).

## Simulations demonstrate competing effects of size and protein–particle attraction

To further elucidate the experimental observations, we conducted molecular dynamics simulations of a simple binary system of Lennard–Jones (LJ) particles to probe the effects of client size and interaction with proteins. We simulate a condensed phase of "protein" particles at conditions where phase separation is observed, i.e., below the critical temperature and above the saturation concentration (Supplementary Figs. 4 and 5). We then add "bead" particles to the system to represent large client molecules such as dextrans,

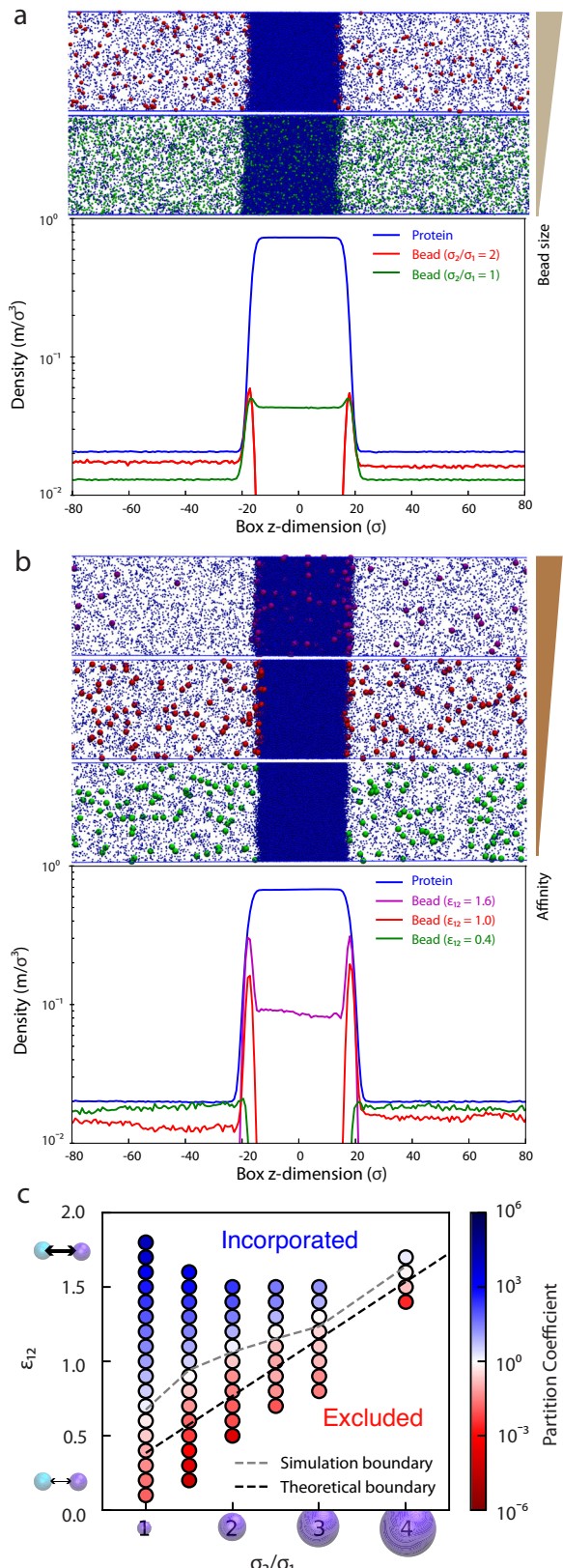

**Fig. 2 | Simulations demonstrate the effect of size disparity and interaction energies on partitioning. a** Density profile of "protein" and "bead" components in slab simulations of Lennard–Jones particles demonstrating the effect of bead size on partitioning. Density refers to reduced mass per unit volume. Smaller beads incorporate most easily, while larger beads are excluded. The inset shows snapshots from simulations. **b** Density profile of "protein" component and "bead" component in slab simulations, showing that stronger protein-bead interactions result in the incorporation of beads into the condensate, while weaker protein-bead interactions result in the exclusion of beads. The inset shows snapshots from the simulation. **c** A grid of simulation parameters was tested, demonstrating that partition coefficients increase with decreasing size and with increasing protein-bead interaction energy. The dashed black line shows the crossover point where the partition coefficient is equal to 1 according to the theory derived in Eq. (3), and the dashed gray line shows the crossover point from the simulations. Source data are provided as a Source Data file.

would require a prohibitively large number of protein particles in the simulation. For all cases, we fixed the total volume fraction ratio of protein:bead to 5:1 to be consistent with conditions tested in experiments (Supplementary Table 4).

Using this system, we tested the effects of both the size and stickiness of the beads on partitioning into a condensate by varying two LJ parameters, namely the diameter of the beads, $\sigma_2$, and the protein-bead interaction energy, $\varepsilon_{12}$ (see "Methods"). We first studied the effect of size on partitioning by looking at two cases where beads and protein are placed together in a phase-separating system. In Fig. 2a, we show the density profile of the protein, which forms a dense phase at the center of the box, and a dilute phase outside. Keeping the protein-bead interactions constant at $\varepsilon_{12} = 0.8$ (i.e., 80% of protein–protein interaction strength, $\epsilon_1$, which was set to 1.0), we find that when beads are the same size as the protein particles ($\sigma_2 = \sigma_1 = 1.0$), they are enriched in the dense phase, but when the diameter of the beads is increased to $\sigma_2 = 2.0$ (now 8× the volume of the protein), they are completely excluded from the dense phase. This is analogous to the case of the 10 and 70 kDa dextrans, where the smaller client partitions into the condensate, and the larger is excluded.

The effect of increased size opposing particle partitioning into condensates can be offset, however, by increasing the attractive interactions between the protein particles and beads. In Fig. 2b, we show that for a system of particles of size $\sigma_2 = 3.0$ (27× the volume of protein beads; a disparity similar to ribosome and protein sizes), weak attractive interactions result in full exclusion, while strong attractive interactions result in preferential incorporation. We also find that at intermediate interaction strengths, the beads localize to the surface of the dense phase, but do not partition inside. Similar surface enrichment has been observed previously for proteins and protein clusters at condensate interfaces[16,40,41]. Thus, we largely recapitulate the observations from experiments showing that ribosomes and other large clients such as nanoparticles can be either preferentially incorporated or excluded from condensates, or localized to the surface, and that varying protein-client interaction strength is sufficient to capture each of the differential localizations.

We note that since solvent is not explicitly considered in the simulations, the LJ interactions are purposed to implicitly account for the effect of solvent. One simplification that arises from this is that the protein-bead interactions are equivalent inside and outside the condensate, which may not reflect reality due to the difference in the chemical environment inside the condensate[6,8]. However, this should have minimal impact on the partitioning of beads in the simulation, since the beads form so few interactions with protein molecules in the dilute phase, and the interactions will be effectively modeling the interaction strength of protein-bead interactions inside the condensate. Stronger interactions in the simulation would thus represent beads with more interaction-prone surfaces, or more favorable interactions with particular proteins inside the condensate.

antibodies, or ribosomes. Employing this simplified model allows us to access more significant size disparities between component types than would be otherwise allowed by more complex coarse-grained simulation models. Although we can simulate significant size disparities between different particle types, we still cannot achieve disparities as large as those observed with the protein-PS experiments, as those

We then asked how size and protein-bead attraction compete to control the partitioning of beads into a protein-rich dense phase. In Fig. 2c, we show a grid of conditions tested with varying both $\sigma_2$ and $\varepsilon_{12}$, finding that smaller beads and beads with stronger attractive interactions are more preferentially incorporated into the dense phase, while larger beads and those with weaker attractive interactions are preferentially excluded from the dense phase.

To explain the reason for each of these effects within the simplified LJ model, we formulate an analytical expression (see Supplementary Notes) to represent the energetic component of the transfer free energy (i.e., change in energy of the system upon insertion of a single particle into a dense phase of protein):

$$\Delta U_{transfer} = V_2 U_{(1)} \rho_{(1)} - A_2 \left(\frac{\epsilon_{12}}{\epsilon_1}\right)\left(\frac{U_{(1)}}{A_1}\right) \qquad (1)$$

where $V_2$ and $A_2$ are the volume and surface area of a bead; $U_{(1)}$ and $\rho_{(1)}$ are the average energy per protein particle and average number density, respectively, within a pure condensate of protein particles; and $A_1$ is the surface area of a protein particle. The first term describes the unfavorable energetic penalty of displacing protein particles within a certain volume in the condensed phase having an energy density of $U_{(1)}\rho_{(1)}$. The second term describes the favorable interactions formed between the inserted bead and the condensate-forming protein particles. Expanding the volume and area terms for a spherical bead, we obtain:

$$\Delta U_{transfer} = 4\pi \left(\frac{\sigma_2}{2}\right)^2 U_{(1)} \left[\frac{\rho_{(1)}}{3} \cdot \frac{\sigma_2}{2} - \left(\frac{\epsilon_{12}}{\epsilon_1}\right)\left(\frac{1}{A_1}\right)\right] \qquad (2)$$

By solving for $\Delta U_{transfer} = 0$, we can obtain a linear function to describe the boundary between preferential incorporation and exclusion, assuming negligible entropic contribution to the transfer of free energy. This gives a linear relationship between $\varepsilon_{12}$ and $\sigma_2$ and depends only on the surface area and the pure protein condensed phase density:

$$\epsilon_{12} = \frac{A_1 \rho_{(1)}}{6} \epsilon_1 \sigma_2 = 0.382 \sigma_2 \qquad (3)$$

The density we use in this calculation was obtained by using a case where beads were not incorporated into the condensate (i.e., $\varepsilon_{12} < 0.5$) and fitting the dense phase of protein, yielding $\rho_{(1)} = 0.73$. We have also used the fact that in all the simulations, $\epsilon_1 = 1$ and $\sigma_1 = 1$. This boundary between incorporation and exclusion is shown as a dashed black line in Fig. 2c and shows reasonable agreement with the simulation data, particularly for larger bead sizes. Possible sources of error include the larger concentration of beads in simulations with smaller $\sigma_2$ (elaborated in Supplementary Notes).

From this section, we conclude that an arbitrarily large bead can partition into a condensate provided that the bead's interactions with protein can be made arbitrarily large. We also present evidence that the increase of stickiness needs to be proportional to the radius (not volume) of the particles. Finally, we note that the simulations also suggest an explanation for the enrichment of particles at the condensate surface. Since the protein density $\rho_{(1)}$ decreases along the axis normal to the condensate surface, the energy penalty (first term in Eq. 1) from adding a large particle decreases at the interface, resulting in an enrichment of beads at the interface.

## Partitioning can be controlled through specific binding

Based on the simulations and experimental results so far, we hypothesized that diverse protein-ligand interactions can overcome the thermodynamic penalty that would otherwise exclude large particles from condensates. One of the strongest known non-covalent protein-ligand interactions is between streptavidin and biotin[42]. The pair has an unusually high affinity of $K_d$ ~ $10^{-15}$ M as well as high specificity, and we therefore sought to harness this interaction to control particle partitioning (Fig. 3a). To test this, we fused streptavidin to the RGG domain to create a streptavidin-RGG fusion protein (SA-RGG) (Supplementary Fig. 1). To prepare the particles, we started from the premise that PS beads must first be PEGylated to block non-specific interactions, with biotin then displayed at the free end of the PEG. We therefore prepared PS-PEG-biotin beads. Remarkably, 90% of PS-PEG-biotin beads with 500 nm diameter partitioned into SA-RGG condensates, whereas 85% of the control particles (500 nm PS-PEG) were excluded (Fig. 3b and Supplementary Fig. 6). The result was the same whether the beads were added before or after the condensates formed (Supplementary Fig. 7). These results indicate that specific, high-affinity interactions can drive the thermodynamic partitioning of large particles into biomolecular condensates.

Interestingly, when the PS-PEG-biotin particles were prepared (see "Methods") with reduced biotin surface density (15%), the particles predominantly adsorbed to the SA-RGG condensate interface (Fig. 3b). This agrees with the simulation results, in which beads of intermediate interaction strength localized at the condensate periphery rather than inside or outside the condensates.

To further explore the interplay of size and affinity, we prepared various PS-PEG-biotin particle sizes and analyzed partitioning into SA-RGG condensates. To avoid confounding variables, the biotin surface density was kept consistent across the different-sized particles, at an intermediate biotin density (0.02 biotin/nm²). We observed that the partition fraction decreased with increasing bead size (Fig. 3c): 100 nm beads predominantly partitioned into the condensates, whereas 200 nm beads were partially inside and partially at the interface, and 500 nm beads were predominantly localized to the condensate interface. This result demonstrates that larger size indeed hinders the partitioning of particles into condensates, given consistent ligand area density on the particles. This agrees with the simulations at constant surface interaction strengths (Fig. 2a), where larger particles were excluded from the condensed phase despite having greater surface area and potential to interact with larger numbers of proteins. This is explained by the derived analytical theory, in which the attractive force driving the inclusion of beads scales with bead surface area, while the excluding force arising from displacement of protein–protein interactions scales with bead volume.

## Protein-nucleic acid interactions can drive particle partitioning

Our experiments with ribosomes (Fig. 1b) suggest that protein-nucleic acid interactions can drive particle partitioning into condensates. We therefore asked whether these interactions can be harnessed to engineer particles for controlled partitioning. To test this, we again began with PS-PEG particles, which resist protein adhesion and are excluded from condensates (Supplementary Fig. 8). We further modified the PS-PEG beads by conjugating DNA oligonucleotides to the free end of the PEG using strain-promoted azide-alkyne cycloaddition (SPAAC) click chemistry (Fig. 4a). We first studied polyA20 conjugated to 500 nm PS-PEG (PS-PEG-polyA20). We observed that these PS-PEG-polyA20 beads partitioned robustly into N protein and (GRGNSPYS)$_{25}$ condensates but adsorbed to the surface of LAF-1 RGG condensates (Fig. 4b), all in 150 mM NaCl buffer. Interestingly, the PS-PEG-polyA20 bead partitioning displayed protein sequence dependence that qualitatively agrees with the trends we observed for ribosome partitioning, presumably because both are determined by condensate-nucleic acid interactions.

Previously (Figs. 1a–c and 4b), N protein was mixed with polyA RNA to induce its phase separation. We hypothesized that there could be competing effects between PS-PEG-polyA and polyA RNA for binding to N protein. To test this, we compared the partition coefficient of PS-PEG-polyA20 particles into N protein condensates with

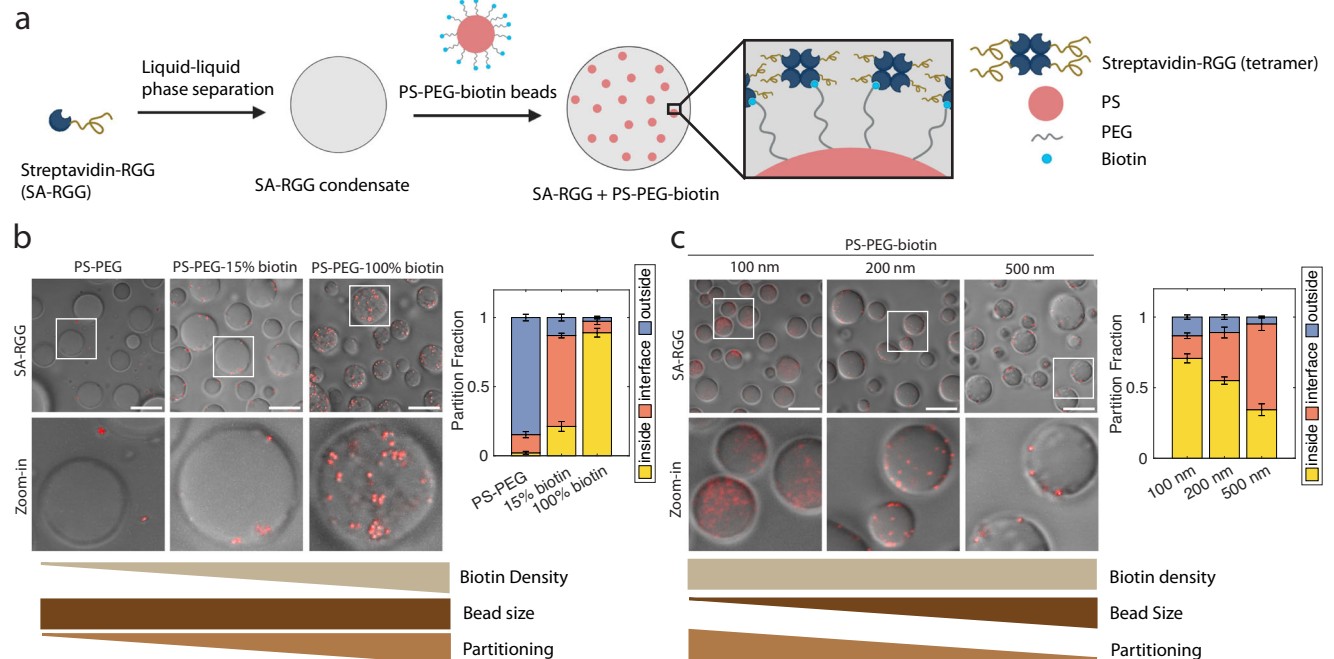

**Fig. 3 | Streptavidin-biotin interactions drive bead partitioning into condensates. a** Schematic model of PS-PEG-biotin partitioning into SA-RGG condensates. Streptavidin (SA, which tetramerizes) binds biotin, resulting in bead recruitment into condensates. Created in BioRender. Kelley, F. (2025) https://BioRender.com/o27h472. **b** Increasing biotin surface density on particles leads to increased partitioning of beads into SA-RGG. (Right) Partition fractions were quantified for each bead. **c** When biotin surface density is held constant, smaller beads display higher partitioning. (Right) Partition fractions were quantified for each bead. Scale bars, 10 μm. Error bars represent SEM with n = 10 from 2 independent experiments. Source data are provided as a Source Data file.

varying polyA RNA concentrations: no RNA, 1× RNA (0.5 mg/mL, the RNA concentration used in our prior N protein experiments), or 2× RNA (1 mg/mL). In the case of no RNA, 8 kDa PEG was used as a crowding agent to induce phase separation of N protein[5]. The absence of RNA led to more robust partitioning of PS-PEG-polyA beads into condensates, whereas 2× RNA concentration resulted in >98% of PS-PEG-polyA beads being excluded from the condensates (Fig. 4c). Motivated by this finding, we revisited the ribosome experiment, where we had previously observed inhomogeneous partitioning of ribosomes into N protein + 1× RNA condensates (Fig. 1b). We now tested ribosome partitioning into N protein condensates lacking RNA and using 8 kDa PEG as a crowding agent, resulting in robust and homogeneous partitioning of ribosomes in the condensates (Supplementary Fig. 9). This result demonstrates that "free" nucleic acids can compete with particle-conjugated nucleic acids for binding to N protein, and hints at a possible mechanism by which condensates exclude unwanted clients in cells.

Since electrostatic attraction between polyA20 and condensates likely plays an important role in PS-PEG-polyA20 partitioning, we hypothesized that varying salt concentration will alter partitioning, as cations in solution will screen the negatively charged oligonucleotides. To test this, we compared the partitioning of the beads at different NaCl concentrations (Fig. 4d). Whereas most PS-PEG-polyA20 beads adsorbed at the interface of RGG condensates in 150 mM NaCl buffer, the beads partitioned further into the condensates in 50 mM NaCl buffer. Upon fluorescently staining the condensates, we observed that at 150 mM NaCl, the interfacial beads are partially immersed in the RGG phase and partially immersed in the dilute aqueous phase. In contrast, at 50 mM NaCl, even beads near the interface are entirely or almost entirely immersed in the RGG phase, indicating that the bead-condensate interfacial tension decreases with NaCl concentration[43] (our resolution is insufficient for accurate contact angle measurements). Similarly, PS-PEG-polyA20 beads partitioned robustly into (GRGNSPYS)25 condensates at lower salt

concentrations, whereas they remained localized at the condensate interface at higher salt concentrations (Supplementary Fig. 10). (Low salt concentration promotes phase separation of LAF-1 RGG, but high salt concentration promotes phase separation of (GRGNSPYS)25; we selected suitable buffer conditions for phase separation based on prior characterization of these proteins[17,26]. Despite the different phase behavior of these two proteins, in both cases, lower salt concentration favored PS-PEG-polyA20 particle partitioning.) We conclude that electrostatic interactions play an important role in driving the partitioning of PS-PEG-polyA20 beads into these condensates.

We conducted additional experiments to verify our results with PS-PEG-polyA20 beads. Orthogonal projections and three-dimensional renderings from confocal microscopy confirm the inclusion of PS-PEG-polyA20 and the exclusion of control beads from condensates (Supplementary Fig. 11). We tested beads of a different size (200 nm PS-PEG-polyA20), and separately, we tested beads with a different oligonucleotide sequence (PS-PEG-polyT20). Both partitioned into condensates, to different degrees (Supplementary Fig. 12). Beads of various sizes conjugated with polyA20 hybridized to polyT20 oligonucleotides (PS-PEG-polyA20/T20) also partitioned into condensates (Supplementary Fig. 13). Overall, these results demonstrate that large particles coated with single or double-stranded oligonucleotides can partition into biomolecular condensates, provided that the condensate-nucleic acid interactions are sufficiently strong.

To further develop this idea, we next asked whether the density of oligonucleotide on the particle surface tunes partitioning. We prepared beads functionalized with varying surface densities of polyA20 (0%, 10%, 75%, and 100%; see "Methods"). We mixed these beads with LAF-1 RGG in 50 mM NaCl buffer and observed that increasing oligonucleotide surface density increased partitioning (Fig. 4e and Supplementary Fig. 14). Similarly, we asked whether particle partitioning could be tuned based on oligonucleotide length. We therefore compared PS-PEG-polyA beads with different polyA lengths: PS-PEG-polyA5, PS-PEG-polyA20, and PS-PEG-polyA40. We tested the

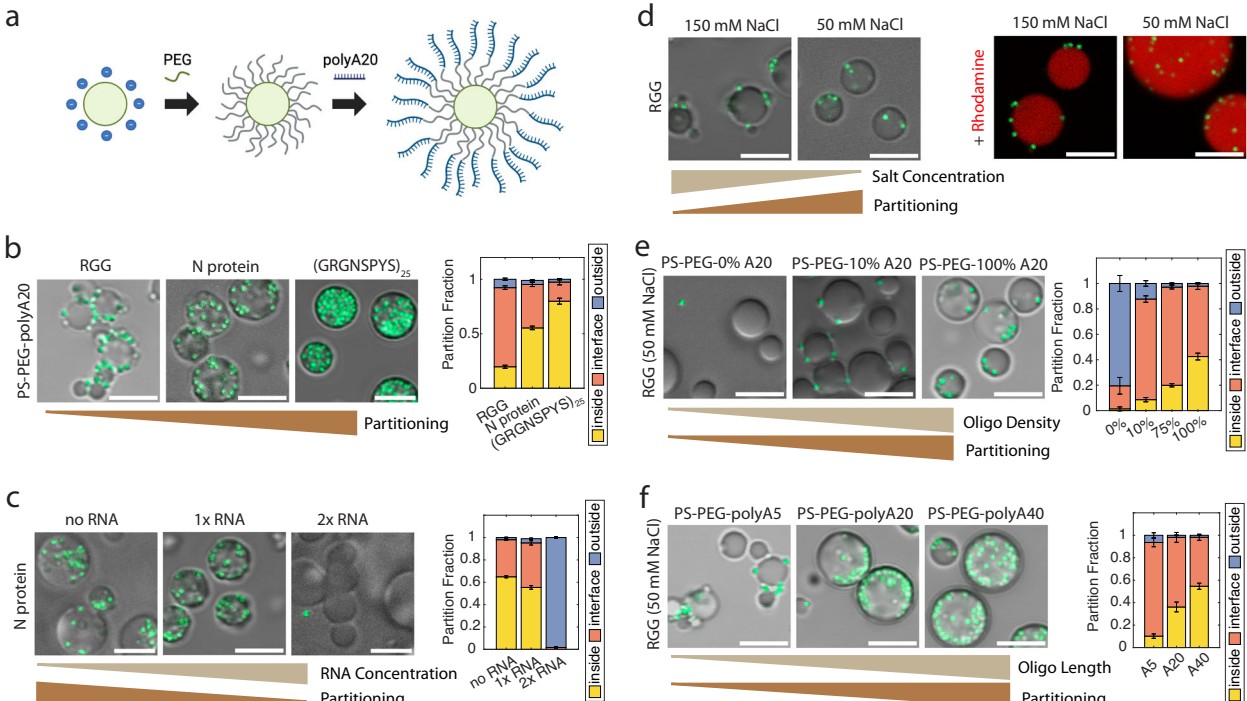

**Fig. 4 | PS-PEG-oligonucleotide beads partition into condensates based on electrostatic interactions. a** Schematic of PS particle surface functionalization to produce PS-PEG and then PS-PEG-polyA20. Created in BioRender. Kelley, F. (2025) https://BioRender.com/o27h472. **b** Proteins were mixed with 500 nm PS-PEG-polyA20 (green) beads. Particles mainly stick to the interface of RGG condensates, but partition more robustly into N protein and (GRGNSPYS)$_{25}$ condensates. (Right) Partition fractions were quantified for each sample. **c** Increasing RNA concentration leads to decreased partitioning of PS-PEG-polyA20 into N protein condensates. (Right) Bead partition fractions were quantified at each RNA concentration. **d** 500 nm PS-PEG-polyA20 beads partition into RGG more at lower salt

concentration, and less at higher salt. (Right) Rhodamine was added to better visualize particle location with respect to the condensate interface. **e** Partitioning of 500 nm beads with 0, 10, and 100% oligo density. Increased oligonucleotide surface density on beads leads to greater partitioning into condensates. (Right) Partition fractions were quantified for 0, 10, 75, and 100% oligo density. **f** PS-PEG-polyA5, A20, and A40 partitioning into RGG condensates at 50 mM NaCl. Longer oligonucleotides lead to higher partitioning into condensates. (Right) Partition fractions were quantified for each bead. Scale bars, 5 μm. Error bars represent SEM with $n \geq 10$ images from at least two independent experiments. Source data are provided as a Source Data file.

partitioning of these beads into RGG in 50 mM NaCl buffer. Indeed, increasing oligonucleotide length resulted in increased partitioning (Fig. 4f). PS-PEG-polyA5 beads were predominantly localized at the condensate interface, whereas nearly 40% of PS-PEG-polyA20 beads and 60% of PS-PEG-polyA40 beads partitioned into the condensates. Together, these results suggest that the strength of condensate-nucleic acid interaction determines PS-PEG-polyA particle partitioning and can be tuned by polyA attachment density or length.

### Reversing particle partitioning

So far, we have demonstrated how nanoparticle surface chemistry can be engineered to drive particle partitioning into condensates. Next, we asked whether condensate-particle interactions can be blocked to prevent partitioning or to expel partitioned particles. Building on Fig. 4c, we sought to understand whether competition between free and particle-bound ligands depends on which species binds to the condensate first. When we mixed LAF-1 RGG condensates (at 50 mM NaCl) with free Cy5-labeled polyA20 (i.e., DNA strands that are not attached to particles), and then subsequently added PS-PEG-polyA20 beads, we observed that the oligos partitioned into the condensates whereas the beads were excluded (adsorbed to the condensate interface)−both when observed after 10 min and after 24 h (Fig. 5a). Similarly, when SA-RGG condensates were mixed with free fluorescently labeled biotin (biotin-4-fluorescein) before adding in PS-PEG-biotin beads, the free biotin partitioned but the beads were excluded−both when observed after 10 min or 24 h. Thus, in both systems, adding abundant free ligands first can occupy binding sites and prevent the partitioning of beads. However, a difference was observed when

particles were added first, before the free ligand (Fig. 5b). When we first allowed the PS-PEG-polyA20 beads to partition into RGG condensates before adding free Cy5-polyA20, after 10 min of equilibration the beads remained partitioned inside even while the oligos also partitioned in, but after 24 h, the beads were predominantly excluded (adsorbed to the interface) of similarly sized droplets. In contrast, PS-PEG-biotin beads that had partitioned into SA-RGG condensates remained partitioned even after the subsequent addition of biotin-4-fluorescein−both when observed after 10 min and after 24 h. We hypothesize that equilibrium partitioning is achieved for PS-PEG-polyA20 beads within 24 h, resulting in free ligand displacing the beads, whereas the PS-PEG-biotin beads are kinetically trapped due to the long lifetime of biotin-streptavidin interactions[44].

This motivated us to ask whether directly weakening the particle-condensate interaction could expel partitioned beads. We have seen that PS-PEG-polyA20 partitioning is salt concentration-dependent (Fig. 4d), so we hypothesized that once the beads are partitioned in, raising the salt concentration may expel the beads from the condensates. We conducted this experiment with PS-PEG-polyA20 particles partitioned into RGG condensates at 50 mM NaCl. Raising the salt concentration from 50 to 150 mM NaCl triggered immediate and rapid transport of PS-PEG-polyA20 outward from the condensates (Fig. 5c). After equilibration, the sample (now at 150 mM NaCl) displayed the same droplet morphology and particle partitioning as samples freshly prepared with the same salt concentration (Supplementary Fig. 15 compared to Fig. 4d). This result suggests that increasing the salt concentration has a rapid and drastic effect on the energetic landscape inside the condensates.

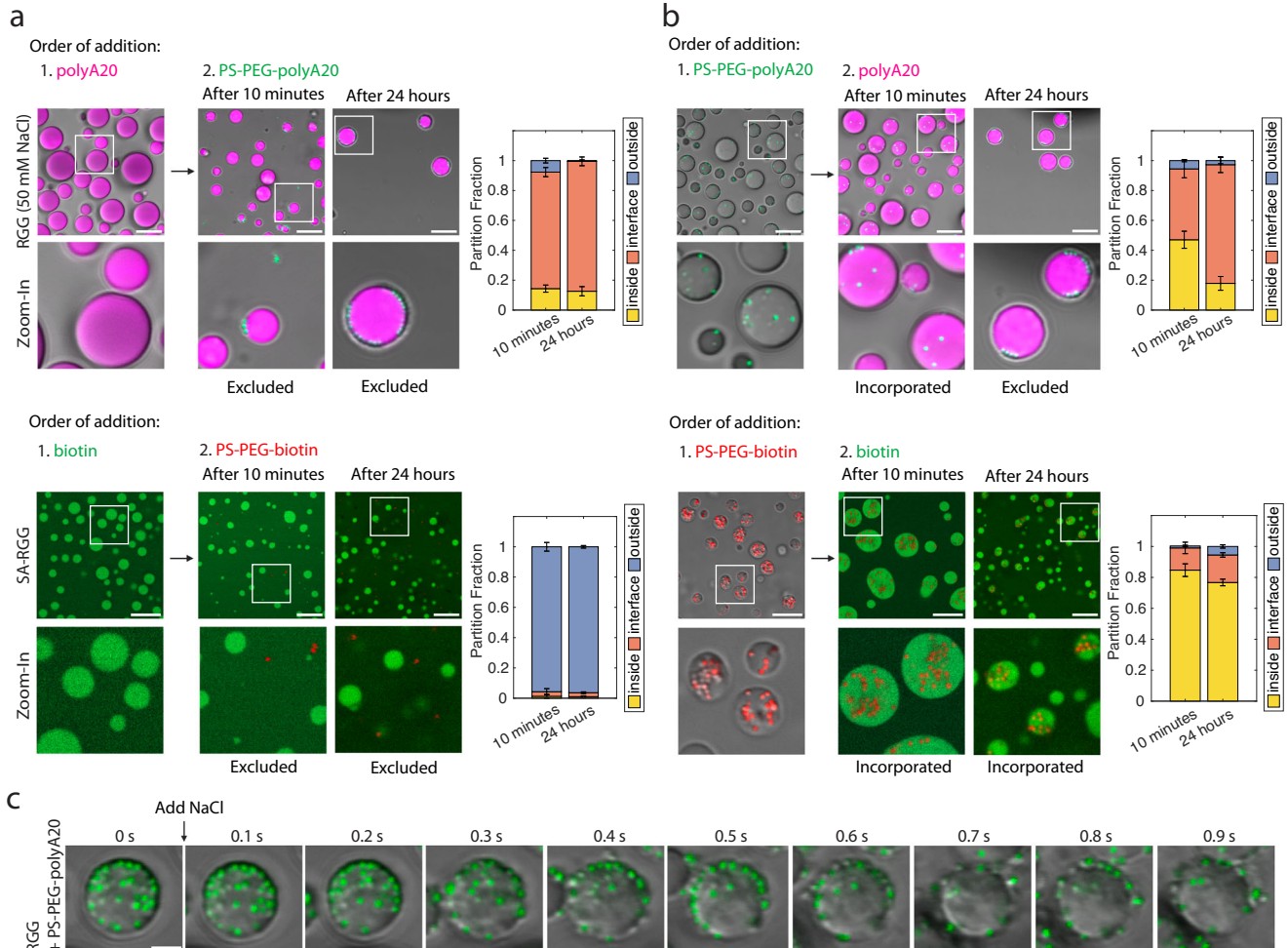

**Fig. 5 | Expulsion of beads from condensates. a** Free ligands prevent beads from partitioning into condensates. (Top) Free Cy5-polyA20 prevents PS-PEG-polyA20 (green) from partitioning into RGG condensates. (Bottom) Biotin-4-fluorescein prevents PS-PEG-biotin (red) from partitioning into SA-RGG (Scale bars, 10 μm.). (Right) Partition fractions were quantified. Data represent mean values and error bars represent SEM with $n \geq 10$ images from 2 independent experiments. **b** Kinetic trapping of beads in condensates. (Top) When Cy5-polyA20 is added after PS-PEG-polyA20 is already partitioned in, beads stay inside condensates after 10 min, but are excluded after 24 h equilibration. (Bottom) Adding biotin-4-fluorescein after PS-PEG-biotin already partitioned results in beads remaining inside condensates, even at 24 h. (Right) Partition fractions were quantified. Error bars represent SEM with $n \geq 8$ images from 2 independent experiments. **c** Time-lapse images of beads being expelled from condensates. RGG with PS-PEG-polyA20 (green) was originally at 50 mM NaCl. Salt concentration was raised to 150 mM NaCl at $t = 0$. Beads rapidly transport out of the condensates. Experiment was repeated independently three times, with similar results. (Scale bar, 5 μm.) Source data are provided as a Source Data file.

## Orthogonal particle partitioning

We have established that distinct protein-ligand interactions can drive particles to partition into various condensates and that the strength of these interactions is tunable. We were curious whether these features permit selective partitioning. We first asked whether the interface and interior of condensates could be simultaneously targeted by tuning interaction strength. Indeed, when we mixed both PS-PEG-polyA5 and PS-PEG-polyA40 beads with LAF-1 RGG, we observed that PS-PEG-polyA40 beads predominantly partitioned into the condensates, while PS-PEG-polyA5 adsorbed to the interface (Fig. 6a). Similarly, PS-PEG-100%biotin beads partitioned into SA-RGG condensates while simultaneously PS-PEG-10%biotin was localized at the interface, due to the beads' high (100%) vs. low (10%) biotin surface density. This difference is highlighted by a radial density profile of the beads (Fig. 6a). Similar results were observed when RGG was mixed with both PS-PEG-10% polyA20 and PS-PEG-100%polyA20 (Supplementary Fig. 16).

We next tested how beads with distinct surface chemistries would partition when combined. We, therefore, mixed PS-PEG-polyA20 and PS-PEG-biotin particles with SA-RGG, and separately, with N protein. In line with our previous results, PS-PEG-biotin

partitioned robustly into SA-RGG condensates (in 150 mM NaCl buffer), while PS-PEG-polyA20 remained at the periphery (Fig. 6b). Conversely, PS-PEG-polyA20 partitioned robustly into N protein condensates, while PS-PEG-biotin beads were excluded. We conclude that distinct surface chemistries can target particles to distinct condensate microenvironments.

Building upon this result, we asked whether orthogonality of bead-condensate interactions can be observed when SA-RGG and N protein condensates are combined. To test this, we mixed SA-RGG with labeled N protein and added both green PS-PEG-biotin and red PS-PEG-polyA20 beads. Strikingly, SA-RGG and N protein formed immiscible and distinct condensates. PS-PEG-biotin beads partitioned overwhelmingly (>99%) into the SA-RGG condensates, whereas PS-PEG-polyA20 partitioned overwhelmingly (>97%) into the N protein condensates (Fig. 6c). Consistent results were obtained with green PS-PEG-polyA20 beads and red PS-PEG-biotin beads, confirming that the orthogonal partitioning was due to which biomolecules are decorated on the particle surface, and not due to any property of the particle core, such as interactions involving the fluorescent dyes. These results demonstrate that large particles can orthogonally target distinct

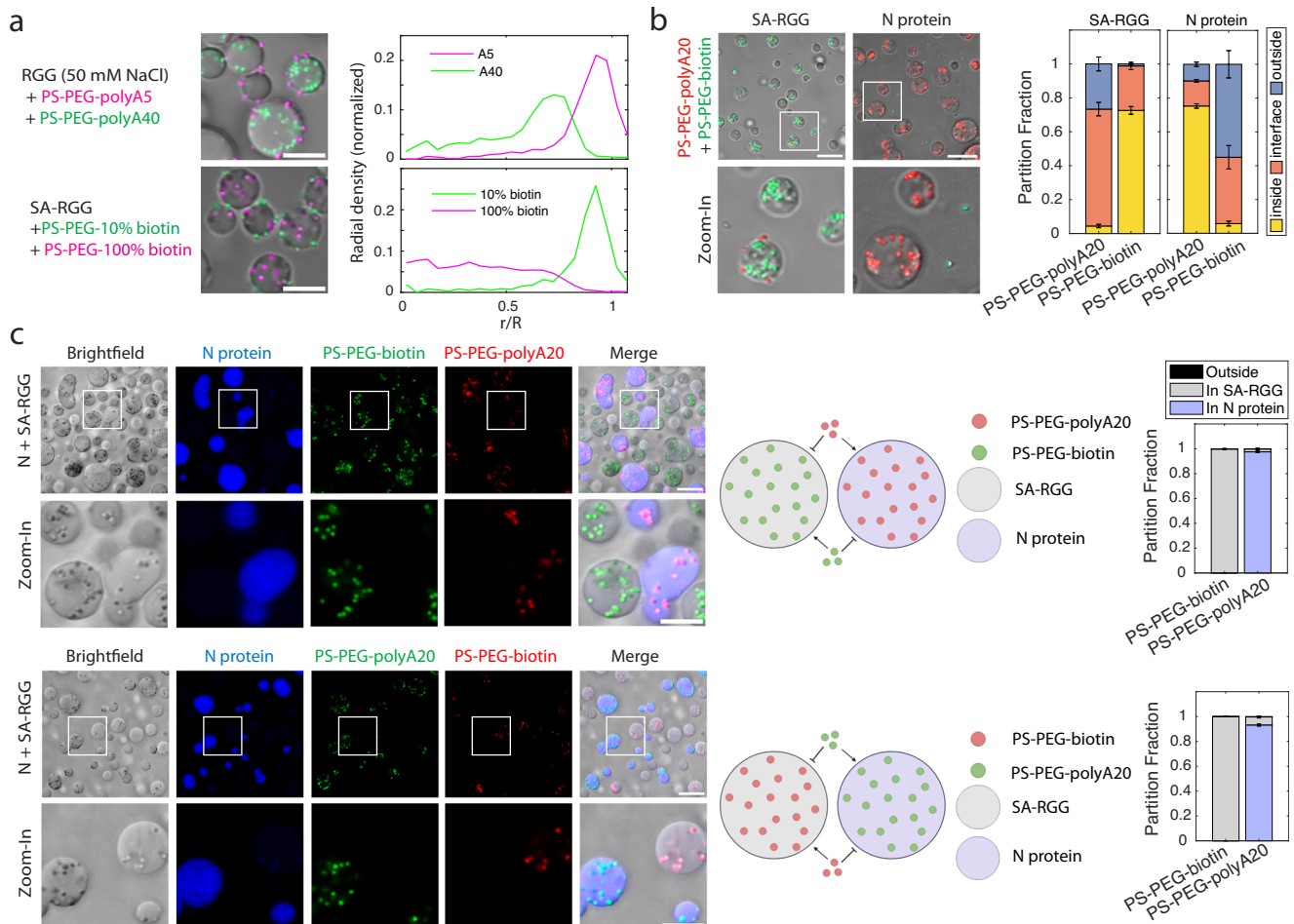

**Fig. 6 | Orthogonality of bead partitioning into condensates. a** Selective partitioning based on condensate-particle interaction strength. PS-PEG-polyA5 (magenta) adsorbs to the condensate interface, while PS-PEG-polyA40 (green) partitions into RGG condensates. Likewise, PS-PEG-10% biotin (green) adsorbs to condensate interface, while PS-PEG-100% biotin (magenta) partitions. (Right) The radial density profile shows bead concentration as a function of normalized distance from the condensate center. **b** Orthogonality of PS-PEG-biotin and PS-PEG-polyA20 partitioning into SA-RGG and N protein condensates. PS-PEG-biotin (green) partitions into SA-RGG, but not into N protein. PS-PEG-polyA20 (red) partitions into N protein, but not SA-RGG condensates. (Right) Partition fractions were

quantified for each bead. Error bars represent SEM with $n = 15$ images from 2 independent experiments. **c** When SA-RGG is mixed with AF647-labeled N protein, they phase separate into immiscible condensates. PS-PEG-polyA20 only partitions into N protein condensates, while PS-PEG-biotin only partitions into SA-RGG. (Middle) Schematic of PS-PEG-polyA20 partitioning into AF647-labeled N protein and PS-PEG-biotin partitioning into unlabeled SA-RGG. Schematic created in BioRender. Kelley, F. (2025) https://BioRender.com/o27h472. (Right) Partition fraction quantified. Error bars represent SEM with $n = 10$ images from 2 independent experiments. Scale bars, 10 μm. Source data are provided as a Source Data file.

condensates in a multi-component system, based on selective particle-condensate interactions.

## Discussion

In this paper, we hypothesized that strong client-scaffold interactions can enable large particles to overcome thermodynamic barriers and partition into liquid condensates. To test this hypothesis, we engineered a toolbox of nanoparticles with diverse sizes and surface chemistries. We found that PEG-coated beads resist partitioning into condensates. In contrast, biotin-functionalized beads partition into streptavidin-tagged condensates based on the high-affinity interaction between biotin and streptavidin, and oligonucleotide-conjugated beads can partition into condensates via protein-nucleic acid interactions—even for beads as large as 1 μm. Partitioning was tuned by modifying oligonucleotide length and surface density, or biotin density, thereby modifying the interaction strength between bead and condensate.

This work expands our understanding of "who's in and who's out" of biomolecular condensates in two significant ways[4]. One key

biophysical insight of our experiments and theory is that arbitrarily large particles can partition into liquid condensates, provided that condensate-particle interactions are sufficiently strong. A second key insight is that large particles functionalized with orthogonal surface chemistries can selectively target immiscible condensates. Previous studies have shown that small molecules with favorable physico-chemical properties partition into condensates[6]. Previous studies have also demonstrated that larger (~5 nm), "inert" molecules are excluded from condensates, whereas proteins and nucleic acids that interact with condensates can partition[45]. However, it was unknown whether larger particles can be both controllably and orthogonally targeted to biomolecular condensates. Our work demonstrates that the answer is yes, with important implications for biology and bioengineering.

With respect to condensate biology, our study may provide insights into how partitioning is spatiotemporally regulated in cellular condensates. Many large biomolecular complexes are believed to assemble and/or function in condensates (e.g., ribosomes, RNA PolII, spliceosomes)[46,47,48], and our study quantifies the tradeoff between size, adhesiveness, and partitioning of such complexes. For instance,

our work is consistent with the proposed mechanism of vectorial flux of ribosomal precursors in the nucleolus[49]. In this model, relatively nascent rRNA transcripts interact more strongly with scaffold components (such as NPM1 and SURF6) in the granular component of the nucleolus. However, as the transcripts mature and bind to ribosomal proteins, the number of multivalent binding sites for scaffold proteins is reduced, thereby thermodynamically driving the expulsion of fully assembled pre-ribosomal particles from the nucleolus. Our work is also consistent with the finding that large cargo, such as the HIV-1 capsid, may be transported through the NPC based on adhesive interactions with FG domains that, by competition, "melt" inter-FG-repeat interactions[14,15]. In synthetic biology, it has been demonstrated that synthetic organelles can be hubs for specific and selective protein translation[50–52]. Our study implies that a key to the design of condensates for orthogonal protein translation is engineering robust recruitment of ribosomes into the synthetic condensates.

Our study focused on particles functionalized with biotin or polyA and examined their partitioning into three model condensates in vitro, but we expect that the approach and principles are generalizable. Besides those tested here, many other small molecule ligands and biopolymers could be used to target particles into a wide variety of condensates, even in more complex biological condensates. For instance, a logical extension is to functionalize particles with oligonucleotides or peptides of a defined sequence to explore sequence-specific partitioning. The diversity of possible particle functionalities suggests exciting possibilities for drug delivery. Some of the main challenges in drug delivery are limited penetration to the targeted microenvironment[53] and non-specific distribution[54]. To address these challenges, modification of nanoparticle size and surface properties, such as PEGylation and tissue targeting moieties, have been extensively explored in nanoparticle-based drug delivery[55]. Our work suggests that similar approaches can be extended to drug delivery to cellular condensates. Our experiments demonstrated orthogonal, specific, and efficient partitioning of particles into condensates, so it may be possible to engineer particles to target disease-related biomolecular condensates, such as in neurodegenerative diseases[56], cancer[57,58], and viral infection[59].

Our experiments, simulations, and theory focused on the thermodynamics of partitioning and do not directly explain the molecular details of how particles interact with condensates, nor how the particles diffuse into the condensates. However, related studies provide insights. Given that the hydrodynamic diameter of our particles significantly exceeds the mesh size of condensates (as measured by inert probes), particle diffusion into and within the condensates is unlikely to involve hopping between mesh cages[60–62]. Instead, we speculate that on timescales longer than the characteristic relaxation times of the biopolymers within the condensates, these biopolymer chains diffuse and rearrange around the particles, potentially opening pathways for the particles to pass through. Indeed, intrinsically disordered proteins remain highly dynamic within condensates, with chain reconfiguration and exchange of interaction partners occurring on the sub-microsecond timescale[63]. An additional mechanism may be at play in some cases: the particles may compete for the same interactions that drive phase separation, which may locally open the condensate and allow the particle to become incorporated into the condensate. This would not likely occur for SA-RGG condensates with PS-PEG-biotin particles, because the particles bind to the proteins at domains not directly involved in driving phase separation, but it may well be relevant for the N protein + RNA condensates with PS-PEG-polyA particles. As mentioned previously, a similar mechanism has been proposed to explain the translocation of nuclear transport receptors and cargo across the NPC, which is itself hypothesized to be a dense protein phase composed of intrinsically disordered proteins[64–66].

There are numerous biomolecular condensates in cells, having different compositions and carrying out various crucial roles. How do these condensates regulate what goes in and out? How can we leverage these properties for biotechnology applications? Our work suggests there is no size limit to partitioning and demonstrates orthogonal partitioning of particles into condensates. The principles revealed in this paper can serve as a foundation to help elucidate how biological condensates regulate partitioning and can be harnessed for therapeutically targeting condensates.

## Methods
### Cloning
RGG, N protein, and MBP-SA-RGG were cloned into a pET vector in-frame with 6xHis-tag using NEBuilder HiFi DNA Assembly (New England BioLabs) and appropriate primers and PCR products or synthetic gene fragments (gBlocks; IDT). Gene sequences were verified using whole plasmid, long-read sequencing (Plasmid-EZ, GENEWIZ). (GRGNSPYS)$_{25}$ was cloned by Genscript into a pQE80L vector. Plasmids will be made available on Addgene.

### Protein expression and purification
RGG, N protein, and MBP-SA-RGG were expressed and purified using methods previously described[5,22,41]. Starter culture was prepared by inoculating a single colony of BL21 (DE3) competent *E. coli* in 5 mL sterile LB media with 50 μg/mL kanamycin and grown overnight. The culture was used to inoculate 500 mL of TB media (Fisher Scientific) supplemented with 4 g/L glycerol and 50 μg/mL kanamycin and grown in a 37 °C shaker at 250 rpm. To induce protein expression, isopropyl β-D-1-thiogalactopyranoside (IPTG) was added to a final concentration of 500 μM when OD600 reached 0.7−1.0, and grown for 18 h at 18 °C. The cell pellet was harvested by centrifugation at 4200 rpm for 15 min at 4 °C and resuspended in lysis buffer (1 M NaCl, 20 mM Tris, 20 mM imidazole, Roche EDTA-free protease inhibitor, pH 7.5), then sonicated on ice. After centrifugation at 20,000 rpm for 30 min at 37 °C, the supernatant was filtered (0.22 μm) and purified through FPLC (AKTA Pure) with a 1 mL nickel-charged HisTrap column (Cytiva). Proteins were washed with 500 mM NaCl, 20 mM Tris-HCl, 20 mM imidazole, pH 7.5 buffer and eluted with 500 mM NaCl, 20 mM Tris-HCl, 500 mM imidazole, pH 7.5 buffer. N protein was purified in a similar method, but proteins bound to the column were additionally washed with 3 M NaCl, 20 mM Tris-HCl, 20 mM imidazole, and pH 7.5 buffer prior to elution. Only eluted N protein fractions with A260/A280 ratio <0.7 were used, to avoid DNA and RNA contamination.

Purified RGG, MBP-SA-RGG, and N proteins were dialyzed overnight using 7 kDa MWCO membranes (Slide-A-Lyzer G2, Thermo-Fisher) in appropriate buffers. RGG was dialyzed in 150 mM NaCl, 20 mM Tris, pH 7.5, at 45 °C to inhibit phase separation. MBP-SA-RGG was dialyzed at room temperature and sterile-filtered through a 0.45 μm filter (SLHPX13NL; MilliporeSigma). N protein aliquots with A260/A280 ratio <0.7 were dialyzed into 300 mM NaCl, 20 mM Tris buffer, pH 7.5 at room temperature. Dialyzed RGG and N protein aliquots were flash frozen and stored at −80 °C. Dialyzed MBP-SA-RGG aliquots were stored at 4 °C.

GRGNSPYS was transformed into *E. coli* M15-[pREP4] strain and recombinantly expressed and purified using previously described methods[26]. The starter culture was prepared by inoculating a single colony of *E. coli* M15-[pREP4] in 150 mL of sterile LB media with 100 μg mL$^{-1}$ ampicillin and grown overnight. The culture was used to evenly inoculate 6 × 750 mL of media (yeast extract 10 g L$^{-1}$, NaCl 5 g L$^{-1}$, and tryptone 16 g L$^{-1}$) and grown in a 37 °C shaker. To induce protein expression, IPTG was added to a final concentration of 1 mM when OD600 reached 0.6−0.8, and grown for 8 h. Cell pellets were harvested by centrifugation at 5000 rpm for 15 min at 4 °C and frozen with liquid nitrogen. For protein purification, cell pellet was thawed and resuspended in pH 8.0 native lysis buffer (50 mM NaH$_2$PO$_4$, 300 mM NaCl, and 10 mM imidazole) with 0.45 g of lysozyme, then sonicated on ice using Fisher Scientific model 500 Sonic

Dismembrator (10 mm tapered horn) for 20 min with 10 s recovery time and subsequently incubated with RNAse (10 µg mL$^{-1}$) and DNAse (5 µg mL$^{-1}$) for 30 min. After centrifugation at 20,000 rpm for 15 min at 4 °C, the pellet was collected and resuspended in denaturing lysis buffer B (8 M urea, 100 mM NaH$_2$PO$_4$, 10 mM Tris·Cl, pH 8.0) via sonication for 1 min with a 10 s recovery time. The supernatant was collected from centrifugation at 20,000 rpm for 15 min at 4 °C, and the pH was adjusted to 8.0, followed by incubation with Ni-NTA resin for 1 h at room temperature. The protein-loaded resin was then loaded into a gravitational flow column, washed with denaturing lysis buffer B, denaturing wash buffer C (8 M urea, 100 mM NaH$_2$PO$_4$, 10 mM Tris·Cl, pH 6.3), denaturing elution buffer D (8 M urea, 100 mM NaH$_2$PO$_4$, 10 mM Tris·Cl, pH 5.9), and eluted with 75 mL denaturing elution buffer E (8 M urea, 100 mM NaH$_2$PO$_4$, 10 mM Tris·Cl, pH 4.5). Elution E fractions were carefully transferred and dialyzed in MWCO 3.5 kDa cassettes against 5 L of deionized water at room temperature with at least 7 changes of water before lyophilization. Lyophilized samples were dissolved in 1x PBS buffer at double the desired concentration and sonicated for 2 min with a 10-s recovery time, followed by filtration (0.45 µm, polyvinylidene fluoride) at high temperature and incubated at 80 °C prior to microscopy.

SDS-PAGE was carried out on all proteins to confirm purity using NuPAGE 4–12% Bis-Tris gels (Invitrogen) followed by incubation with Coomassie stain (GelCode™ Blue Safe Protein Stain; ThermoFisher). Protein concentrations were measured using a NanoDrop One Microvolume Spectrophotometer (ThermoFisher). RGG and GRGNSPYS concentrations were measured in buffer with 4 M urea to prevent phase separation.

## Preparation and storage of purchased proteins and small molecules

Lyophilized polyA RNA (P9403; Sigma) used to induce N protein phase separation was resuspended in MilliQ water to 10 mg/mL, flash frozen, and stored at −80 °C. Antibodies were purchased from ThermoFisher: SARS-CoV-2 Nucleocapsid protein rabbit polyclonal IgG (catalog #: SARS-COV2-N-FITC. Lot #: 3170.IG.20) and Rabbit IgG Isotype Control, FITC (catalog # 11-4614-80. Lot #: 2685487) and stored according to the manufacturer's recommendations. Ribosomes were purchased from New England Biolabs (*E. coli* Ribosome) and fluorescently labeled with DyLight 650 NHS Ester (DL650) (catalog # 62265; ThermoFisher) using a previously described protocol[51]. Labeling reaction was carried out by combining 4.8 µM ribosome, 50 mM Tris-HCl, pH 7.6, 15 mM MgCl$_2$, 100 mM NH$_4$Cl, 6 mM β-mercaptoethanol, 250 µM DyLight650 NHS-ester and incubated for 30 min at 37 °C. After the labeling reaction, ribosomes were thoroughly washed in a centrifugal ultrafiltration device (UFC501096; Sigma) until no fluorescence was observed from condensates mixed with the wash buffer flow-through. Aliquots of labeled ribosomes were flash frozen and stored at −80 °C. Biotin-4-fluorescein was purchased from AAT Bioquest, resuspended in DMSO at 10 mg/mL, and stored at −20 °C. Dextrans of various sizes were purchased from Sigma (R8881, 42874, R9379), resuspended in MilliQ water at 50 mg/mL, and stored at 4 °C.

## Oligonucleotide preparation

Oligonucleotide strands were synthesized using a K&A H-2 synthesizer on Glen UnySupport™ 1000 (Glen Research) with TWIST columns (Glen Research) on a 10 µmol scale. All phosphoramidites and oligonucleotide synthesis reagents were purchased from Glen Research and used as received. The synthesis utilized dA-CE phosphoramidite, dT-CE phosphoramidite, Ac-dC-CE phosphoramidite, and dmf-dG-CE phosphoramidite to incorporate standard ATCG bases. 5′ Cy5 and DBCO modifications were done using Cyanine 5 phosphoramidite and 5′-DBCO-TEG phosphoramidite, respectively. Synthesis was performed using 0.25 M 5-ethylthio-1H-tetrazole (ETT) in anhydrous acetonitrile as the activator and 3% TCA/DCM as the deblocking reagent. The

capping step was carried out by mixing tetrahydrofuran/pyridine/acetic anhydride (Cap Mix A) and 16% 1-methylimidazole in tetrahydrofuran (Cap Mix B). Oxidation was carried out using 0.02 M iodine in tetrahydrofuran/water/pyridine for all non-DBCO-containing strands (with a 40-s oxidation time) and 0.5 M CSO in anhydrous acetonitrile for DBCO-containing strands (with a 3-min oxidation time). Coupling times were set at 55 s for standard ATCG bases, 3 min for Cyanine 5 phosphoramidite, and 10 min for 5′-DBCO-TEG phosphoramidite. Strands were deprotected and cleaved using a 35% ammonia solution (VWR) for 17 h at 55 °C for non-Cy5-containing strands, or 17 h at room temperature for Cy5-containing strands. All oligonucleotides were purified by reverse-phase HPLC (SHIMADZU LC-20AR) using a ZORBAX StableBond 300 C18, 250 × 9.4 mm I.D., 5 µm as the stationary phase, with a flow rate of 3 mL/min, and acetonitrile and 0.06 M triethylamine acetate (TEAA) aqueous buffer as the mobile phase. Once purified, all oligonucleotides were lyophilized and characterized using ESI-HRMS (Waters SYNAPT G2-Si Mass Spectrometry). Purity was confirmed via analytical HPLC using a C18 column (150 × 4.6 mm I.D.) from YMC CO., LTD. A complete list of the synthesized oligonucleotides is shown in Supplementary Table 5.

## Bead preparation

Particle PEGylation reactions were performed using previously described methods optimized by Nance et al.[39]. Sonicated stock PS beads were aliquoted and diluted 2× with MiliQ water in the tube. The diluted beads were sonicated for 7 min. Then, amine-terminated PEG polymers were conjugated to carboxyl-modified PS beads using NHS-EDC chemistry in pH 8.2, 200 mM borate buffer. All PEG polymers were purchased from Creative PEG Works in 5k MW (Supplementary Table 6). Stock reagents were added in the following order: PEG, NHS, borate buffer, EDC. Reagents were added without delay in between to achieve maximum reaction efficiency. All bead modifications were done in 1.5 mL tubes (1615-5500; USA Scientific) prepared by coating with 5% Pluronic F-127 for a minimum of 1 h, followed by rinsing once with 1 mL MilliQ water, to minimize bead loss. Depending on the beads' COOH density (as measured by the manufacturer), the mass of PEG, N-hydroxysulfosuccinimide (sulfo-NHS), 1-ethyl-3-(3-dimethylaminopropyl)carbodiimide hydrochloride (EDC), and the borate buffer volume were determined. Reactions were incubated for a minimum of 4 h at room temperature on a rotary shaker and washed twice by centrifugation. Collected particles were resuspended in the original PS particle stock volume and stored at 4 °C. PEGylation reaction conditions for Fluospheres (ThermoFisher) carboxylate-modified polystyrene microspheres are shown in Supplementary Table 7. PEGylation reaction conditions for fluorescent carboxyl-modified PS beads from Bangs Labs are shown in Supplementary Table 8.

To prepare PS-PEG-biotin beads, biotin-PEG-amine was used. The density of biotin around beads was controlled by mixing biotin-PEG-amine with mPEG-amine in appropriate ratios during PEGylation. For example, to prepare PS-PEG-15% biotin, 15 weight % biotin-PEG-amine was mixed with 85 weight % mPEG-amine. To compare biotin density across different bead sizes, first, the bead concentration was quantified using a plate reader (Molecular Devices, spectraMax M2) in a 96-well black bottom plate (Fisher). Pierce™ Fluorescence Biotin Quantitation Kit (ThermoFisher) was used to determine the biotin density on PS-PEG-biotin beads and measured using the same plate reader. The biotin standard curve was generated using the kit with 488 nm excitation. Bead concentration standard curves were generated for each bead size with unmodified red beads in the plate reader using 561 nm excitation. The bead concentrations of biotinylated red beads were measured and analyzed using the same excitation. Then, the biotinylated beads were analyzed for biotin density using 488 nm excitation. The biotin density for each bead sample was calculated using the 488 and 561 nm standard curves.

To prepare PS-PEG-oligonucleotide beads, the beads were first PEGylated using azide-PEG-amine. Then, an oligonucleotide conjugation reaction was performed through SPAAC click chemistry[67]. DBCO-functionalized oligonucleotides were added to PS-PEG-azide in a 1:1 ratio with respect to the COOH groups in 150 mM NaCl, pH 7.5, and incubated on a rotary shaker for 24 h. The density of oligonucleotide on the beads was controlled beginning in the PEGylation step by mixing azide-PEG-amine with mPEG-amine in appropriate ratios. For example, to prepare PS-PEG-75% polyA20, 75 weight % azide-PEG-amine was mixed with 25 weight % mPEG-amine during the PEGylation reaction. After the reaction was complete, beads were washed twice with MilliQ water by centrifugation. Beads were then resuspended in MilliQ water and stored at 4 °C. Beads were vortexed briefly prior to mixing with proteins for microscopy to prevent bead aggregation.

## Microscopy

Imaging was carried out on a Zeiss LSM900 confocal microscope with an Axio Observer 7 inverted stand and using a 63× plan-apochromatic, oil-immersion objective with 1.4 numerical aperture (NA). Transmitted light images were collected with an ESID module (0.55 NA condenser). Green beads, biotin-4-fluorescein, and fluorescein-labeled antibodies were excited with a 488 nm laser; red beads and rhodamine-labeled dextran with a 561 nm laser; and DL650 labeled ribosomes, AF650 labeled N protein, far-red fluorescent beads, and Cy5-labeled oligonucleotides with a 640 nm laser.

RGG samples were imaged at 1 mg/mL in 150 mM NaCl, 20 mM Tris, pH 7.5, unless otherwise stated. Protein was thawed at 45 °C immediately before imaging to prevent aggregation. N protein was prepared at 45 μM in 150 mM NaCl, 20 mM Tris, pH 7.5 buffer and mixed with 0.5 mg/mL polyA RNA (P9403; Sigma-Aldrich) immediately prior to imaging to induce phase separation. GRGNSPYS was imaged at 18 μM in 1x PBS buffer.

MBP-SA-RGG at 4 mg/mL was reacted with 0.02 mg/mL final TEV protease concentration for 30 min to induce phase separation by removing the MBP tag. This was done in Pluronic F-127 coated microscope dishes. Typically, phase separation was induced before beads were added. For control samples where beads were added before phase separation, beads were added to MBP-SA-RGG before TEV protease treatment.

Beads were added at 0.01 volume % unless otherwise specified. For antibody partitioning experiments, antibodies were mixed with protein to final 0.33 μM. For dextran partitioning experiments, dextrans were mixed with proteins at a final 0.5 g/L. For oligonucleotide partitioning experiments, Cy5-polyA20 was mixed with proteins at a final 5.5 μM. For biotin-4-fluorescein partitioning experiments, biotin-4-fluorescein was mixed with SA-RGG at final 0.1 g/L.

Samples were plated on 16-well glass-bottom dishes (#1.5 glass thickness; Grace BioLabs) after pre-treating the glass with a solution of 5% Pluronic F-127 (Sigma-Aldrich) for a minimum of 10 min and then rinsing with DI water. Plated samples were equilibrated for 10 min before imaging.

## Image analysis

Image analysis and quantification were performed using custom codes in MATLAB R2023b. Condensates were identified in each image by segmentation or by the circular Hough Transform. Bead positions were identified as centroids of bright spots in the appropriate fluorescence channel. The position of fluorescent particles with respect to the condensates was classified as inside, interface, or outside the condensates. For dextrans and antibodies, the partition coefficient was measured as the average fluorescence intensity inside the condensates, divided by the average fluorescence intensity outside the condensates (after subtracting background fluorescence measured from a blank sample). Bar graphs, line scans, and radial concentration profiles were also generated in MATLAB.

## Particle size and zeta potential measurements

Bead size and zeta potential were measured using a Malvern Zetasizer Ultra. Bead size was measured using 1 μL of bead sample diluted into 1 mL MilliQ water or buffer solution in a polystyrene cuvette. Bead zeta potential values were measured using 1 μL of bead sample in 700 μL of buffer in a folded capillary zeta cell (DTS1070; Malvern).

## Molecular dynamics simulations

Simulations were carried out on systems of LJ particles using the LAMMPS simulation engine[68]. In each system, we simulate two types of particles, (1) protein, and (2) an attractive client molecule, simply referred to as a "bead." Protein–protein interactions were handled using a simple LJ functional form:

$$E(r) = 4\epsilon_1 \left[ \left(\frac{\sigma_1}{r}\right)^{12} - \left(\frac{\sigma_1}{r}\right)^6 \right], r < r_{cut} \tag{4}$$

where $\epsilon_1 = 1$ is the depth of the energy well, or the strength of attractive interactions between two protein particles, and $\sigma_1 = 1$ is the characteristic distance of the interactions or the size of the LJ particle representing protein molecules.

Bead-bead interactions were treated as softly repulsive and handled using a Weeks–Chandler–Anderson functional form:

$$E(r) = \begin{cases} 4\epsilon_2 \left[ \left(\frac{\sigma_2}{r}\right)^{12} - \left(\frac{\sigma_2}{r}\right)^6 \right] + \epsilon_2, & r < 2^{\frac{1}{6}}\sigma_2 \\ 0, & r \geq 2^{\frac{1}{6}}\sigma_2 \end{cases} \tag{5}$$

where $\epsilon_2 = 1$ is the scaling of repulsive interactions between beads and $\sigma_2$ is the size of a bead, which is treated as a free variable to test the size-dependence of the partitioning of beads into a condensate of protein particles.

Since the surface chemistry of beads and their interactions with protein should be similar at different sizes, for protein-bead interactions we utilize the LAMMPS LJ-expand force field, which keeps the width of the energy well identical between all bead sizes, and simply shifts it outward further from the center of the bead to increase the bead's size:

$$E(r) = 4\epsilon_{12} \left[ \left(\frac{\sigma_1}{r - \Delta_{12}}\right)^{12} - \left(\frac{\sigma_1}{r - \Delta_{12}}\right)^6 \right], r < r_{cut} + \Delta_{12} \tag{6}$$

where $\Delta_{12} = \frac{\sigma_2 + \sigma_1}{2} - 1$ is the offset value used to shift the LJ functional form outward from the center of the bead and $\epsilon_{12}$ is the interaction energy between protein and beads. For all systems tested, protein particles were assigned default LJ parameters: $\sigma_1 = 1$, $\epsilon_1 = 1$, and mass = 1, while beads were assigned a range of $\epsilon_{12}$ and $\sigma_2$ parameters. Masses of beads were increased to keep density constant at 1, and the scaling of repulsive interactions was set to be similar to that of the protein particles by setting $\epsilon_2 = 1$. Functional forms used for each interaction are shown in Supplementary Fig. 17.

Simulations were conducted using 51200 protein particles, and enough beads to reproduce a 5:1 protein:bead mass ratio. We conducted simulations in slab geometry with a box of size $20 \times 20 \times 200$ $\sigma^3$ for 10 million MD steps using Langevin dynamics at constant temperature: $T^* = 0.9$. The condensed phase was centered in the box using the methodology of Jung et al.[69] and density profiles were calculated by binning the elongated z-dimension into 500 bins and calculating the density of particles at each frame and discarding the first two million steps as equilibration of the system. Partition coefficients were calculated by first calculating the equilibrium concentrations of beads inside and outside the slab. The boundaries of the condensate and external regions were determined by selecting a sub-region of the system where bead concentration is relatively level and averaging values across all bins within this range. Importantly, some systems had

significant interfacial enrichment, and thus the size of the box subregion used for this calculation was smaller in those cases. All analysis was done using Python scripts and utilizing the mdtraj and MDAnalysis libraries.

## Reporting summary

Further information on research design is available in the Nature Portfolio Reporting Summary linked to this article.

## Data availability

Unless otherwise stated, all data supporting the results of this study can be found in the article, supplementary, and source data files. Sample MD simulation trajectories and scripts to set up and run simulations in LAMMPS have been deposited on Zenodo under accession number 14931194. The processed density profiles from simulations and partition coefficient calculations from Fig. 2 are provided in the Source Data file. Source data are provided with this paper.

## Code availability

We conducted all simulations in the open-source LAMMPS MD simulation engine (Updated 22 Dec. 2022). Simulation setup and production are handled through a single script, which is available on Zenodo under accession number 14931194.

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

## Acknowledgements

We thank Cristobal Garcia Garcia, Kristi L. Kiick, Shiv Rekhi, and Jeetain Mittal for designing and providing the (GRGNSPYS)$_{25}$ protein. We also thank Xinyi Li for purifying (GRGNSPYS)$_{25}$ protein. We thank Richard Haber for the use of the Zetasizer. We gratefully acknowledge Elizabeth Nance for the PEGylation protocol. We thank Nina Shapley, Ned Wingreen, and Yaojun Zhang for their helpful discussions. We also thank Srinivas Chakravartula for help with oligonucleotide characterization. This work was supported by NIH grants R35GM142903 (to B.S.) and R35GM150589 (to G.D.). This work was also supported by Rutgers University startup funds (to Y.G. and G.D.) and computing resources through Rutgers Engineering Computing Services. F.K. was supported by NIH training grant T32 GM135141 and B.L. was supported by NSF REU award 2149971. Schematics in Figs. 1e, 3a, 4a and 6c were created in BioRender: Kelley, F. (2025). https://biorender.com/o27h472.

## Author contributions

Fleurie M. Kelley: experimental design, data collection and analysis, writing—original draft, writing—review and editing. Anas Ani: data collection and analysis, writing—review and editing. Emily G. Pinlac: data collection and analysis, writing—review and editing. Bridget Linders: data collection. Bruna Favetta: provision of protein, feedback on the manuscript. Mayur Barai: provision of protein, feedback on the manuscript. Yuchen Ma: provision of oligonucleotides. Arjun Singh: software development, data analysis. Gregory L. Dignon: supervision, software development, data analysis, writing—original draft, writing—review and editing, funding acquisition. Yuwei Gu: supervision, experimental design, provision of oligonucleotides, writing—original draft, writing—review and editing, funding acquisition. Benjamin S. Schuster: conceptualization, supervision, experimental design, software development, data analysis, writing—original draft, writing—review and editing, funding acquisition.

## Competing interests

The authors declare no competing interests.
