## [Transparent Peer Review file · Nature Communications]

Controlled and orthogonal partitioning of large particles into biomolecular condensates

Corresponding Author: Professor Benjamin Schuster

Version 0:

Reviewer comments:

Reviewer #1

(Remarks to the Author)

This is a well-executed study on the engineering of large molecules (cargos) for recruitment to condensates. The manuscript also provides valuable biophysical insights into the factors governing molecule partitioning, particularly focusing on molecule size and binding affinities. The in vitro experiments, simulations, and theory all converge to support the paper's key findings. Overall, the manuscript is well-written, organized, and of high quality, and it should attract significant interest from the community.

However, there are a few areas that require further clarification or elaboration, as outlined below:

1. The comment regarding the diffusion of large molecules in the discussion (line 417) requires further justification or contextualization, especially considering that such behaviors were not directly investigated in the study.
2. Charge appears to be an important factor in ribosome partitioning, but it does not seem to be the only factor at play. This is evident since two of the model systems have the same net charge but exhibit different partitioning behaviors. A more detailed explanation for these differences is warranted.
3. In Fig. 1C, the condensates display unusual "non-spherical shapes." What is the underlying cause of this behavior? Are these systems kinetically arrested?
4. Most of the model systems in this study are positively charged, with cargos recruited via negatively charged interaction sites. While this surface chemistry seems straightforward in this context, it may be more complex for biological condensates. The authors should address how they would approach surface chemistry design in more complex systems. More generally, the limitations of the model systems used in this study should be discussed.
5. For the biotin experiments (line 255), while the surface density remains constant for different bead sizes, the total number of available sites would vary due to the differences in bead size. Did the authors account for this in their experiments? If not, the trends reported in Fig. 3C may be more pronounced than observed.

Reviewer #2

(Remarks to the Author)

Kelley et al study the partitioning of beads inside and around different types of condensates and show that both size and affinity of bead-protein interactions matter to determine the partitioning. This makes sense and it is somewhat obvious. The paper is straightforward and with expected results, nothing particularly surprising or unexpected. The simulations with particles of different sizes and interaction strengths must have been done before in some form. Despite that all this sounds negative, I think it deserves publication in Nat Comm as it is overall quite solid and well evidenced.

I don't find major flaws or poorly evidenced claims.

I can imagine that this paper will be a useful reference for future works on condensates.

I have one important point that needs to be corrected before publication.

For microrheology to work, probe particles need to be NOT interacting with the polymers in solution. For this reason, I suggest the authors to modify their statements accordingly in the conclusions.

Specifically lines 446:

"One is the design of probes for microrheology measurements of condensates. It should be possible to ... condensate rheology".

On this point, maybe the authors will find this review interesting for their current or future work

<https://pubs.acs.org/doi/full/10.1021/jacsau.2c00055>

Typos:

Line 134 "molecules larger than ???? would be ... "

Line 460 "Our experiments demonstrated orthogonal, specific, and efficient partitioning of particles into condensates due to avidity" ? -> do they mean affinity?

Reviewer #3

(Remarks to the Author)

The research of Kelley et al. investigates the partitioning of large particles into biomolecular condensates. Using polymer nanoparticles with tailored surface chemistries, the study demonstrates that strong particle-condensate interactions can drive the inclusion of even large (up to 1 μm) particles into condensates. The partitioning behavior can be tuned by altering factors like ionic strength, length of conjugated ligands, and density of sticky groups at nanoparticle surfaces. Orthogonal recruitment into immiscible condensates is also demonstrated. These findings offer valuable insights into the uptake of large client molecules by model condensates.

The authors present a carefully conducted investigation; experiments were performed with state-of-the-art techniques, and the quality of data and presentation of the results is very good. Their theoretical model satisfactorily recapitulates the experimental behavior.

The quality of experimental data allowed for the conclusive evaluation of how size and surface chemistry influence partitioning efficiency. While the study has undoubtful merits, it also has weaknesses that mitigate its relevance, as outlined below.

1) The manuscript fails to present the key determinants of phase separation for the representative molecules. What is the nature of scaffold-scaffold interactions? What are the main features of the respective condensate environments? Focusing on electrostatic interactions, the authors mention the overall net charges of the three selected molecules but disregard possible sequence or charge patterning contributions. It is unclear what differentiates one condensate from another and their representativeness. It appears that all condensates are driven by (partly) disordered polypeptide chains, while the investigation does not include other types of phase-separating macromolecules such as proteins exhibiting multivalent modular domains. The common and divergent properties of the selected model systems should be better detailed. The presence of co-scaffold molecules (e.g. RNA) should be acknowledged from the very beginning.

2) The theoretical framework proposed to explain the observed behavior provides somewhat limited insight into the chemistry and molecular rules underlying particle partitioning. The model predicts that stronger interactions will counteract particle exclusion of larger particles. Yet, the energetic contributions remain vague and undetailed; solvation effects are apparently not considered, or are they irrelevant? The recruitment of small molecules into certain condensates was described as an emergent property of the separated phase (Thody et al., Nat. Chem. 2024. <https://doi.org/10.1038/s41557-024-01630-w>). Is the recruitment of nanoparticles solely driven by the higher concentration of binding macromolecules within the dense phase (then simply a consequence of mass action), or are there additional, emergent properties at play?

3) The authors intend to demonstrate that larger particles can partition into condensates and that particles can be engineered for controlled partitioning:

"We hypothesized that for other clients, interactions with the scaffold protein can drive partitioning into condensates of clients > 5 nm"

"These observations suggest that particles significantly larger than the dextrans may partition into condensates, depending on condensate-particle interaction strength."

"We therefore asked whether these interactions can be harnessed to engineer particles for controlled partitioning."

The use of large beads or particles to determine properties inside condensates is quite established (although in some cases they are embedded by microinjection) (e.g. Elbaum-Garfinkle et al., PNAS, 2015, <https://doi.org/10.1073/pnas.15048221>; JACS, 2019, Shayegan et al., <https://doi.org/10.1021/jacs.8b13349>; Avni et al., Nat Comm, 2022, <https://doi.org/10.1038/s41467-022-32143-0>; Pan et al., ACS Nano, 2022, <https://dx.doi.org/10.1021/acsnano.0c05486>).

Moreover, there are examples in the literature that describe how surface functionalization with biomolecules can promote the inclusion of nanoparticles into biomolecular condensates driven by interactions (e.g. Barracchia et al., Bioconj. Chem. 2022. <https://doi.org/10.1021/acs.bioconjchem.2c00168>). Accordingly, the text should be modified to acknowledge that this is not a first-time discovery but rather one that advances previous investigations.

4) The presented biomolecular condensates appear stable and unaffected by changes in experimental conditions. For example, high or low ionic strength conditions are not reported to perturb condensate stability. Did the authors test phase behavior in a range of salt concentrations? Supporting phase diagrams could be helpful. Moreover, do the authors have any evidence about the impact of nanoparticle exposure on condensate size and stability?

5) The morphological characterization of the particles used in the experiments needs to be implemented. Hydrodynamic diameter data must be presented with a more appropriate number of significant figures. It is unclear whether the error reported for these measurements is the standard deviation over a bunch of repeated measurements, or if it is the standard deviation of particle size distribution in each sample. In the latter case, certain values appear unrealistic (200 nm PS-PEG-azide 199 ± 0.06 nm; 100 nm PS-PEG-30% biotin 140 ± 0.56 nm; 200 nm PS-PEG-20% biotin 211 ± 0.09 nm). It would be useful to report in Table S1 the polydispersity index values for these measurements to represent the sample size distribution width more clearly. It would also be helpful to show hydrodynamic values under the ionic strength conditions in which the particle samples have been tested (50, 150 mM, and sometimes 900 mM) rather than in pure water. Similar considerations apply to Z-potential measurements. Values should be reported with an appropriate number of significant figures. The same particles are often used in experiments with biomolecular condensates under very different ionic strength conditions (50, 150 mM, sometimes 900 mM). Since Z-potential can be affected by ionic strength, it would be appropriate to show Z potential values under the ionic strength conditions in which the particle samples have been tested and not just in 0.1X PBS buffer.

Minor points:

6) The terminology of the following sentence is specialized: "We reason that this hypothesis will be valid if the condensate is terminally viscous, so that the biopolymer network can flow around the particles at time scales longer than the reptation relaxation time". Please add an explanation.

7) "Absence of RNA led to more robust partitioning of PS-PEG-polyA beads into condensates, whereas 2x RNA concentration resulted in >98% of PS-PEG-polyA beads being excluded from the condensates (Fig. 4C). This result demonstrates that "free" nucleic acids can compete with particle-conjugated nucleic acids for binding to N protein, and hints at a possible mechanism by which condensates exclude unwanted clients in cells." These findings appear obvious, therefore this part is not very informative.

Reviewer #4

(Remarks to the Author)

Version 1:

Reviewer comments:

Reviewer #1

(Remarks to the Author)

The authors have thoroughly addressed the queries raised in the initial review. In particular they provide additional supporting data and clarifications that have served to strengthen the work.

Reviewer #2

(Remarks to the Author)

the authors have addressed my comments

Reviewer #3

(Remarks to the Author)

The authors have addressed all my concerns and made the requested changes. The manuscript can be published.

Reviewer #4

(Remarks to the Author)

Authors' Response to Reviewers' Comments

NCOMMS-24-51553

“Controlled and orthogonal partitioning of large particles into biomolecular condensates”

Kelley et al.

We thank the editor and three reviewers for their valuable feedback on our manuscript and for highlighting the merits of our work. Here and in the revised manuscript, we address the reviewers' major and minor comments about our work. We believe this constructive feedback has allowed us to significantly improve our manuscript.

Reviewer #1:

This is a well-executed study on the engineering of large molecules (cargos) for recruitment to condensates. The manuscript also provides valuable biophysical insights into the factors governing molecule partitioning, particularly focusing on molecule size and binding affinities. The in vitro experiments, simulations, and theory all converge to support the paper's key findings. Overall, the manuscript is well-written, organized, and of high quality, and it should attract significant interest from the community.

We thank the reviewer for highlighting the high quality of our manuscript and noting its significant interest for the research community.

However, there are a few areas that require further clarification or elaboration, as outlined below:

1. The comment regarding the diffusion of large molecules in the discussion (line 417) requires further justification or contextualization, especially considering that such behaviors were not directly investigated in the study.

We thank the reviewer for encouraging us to clarify this point. We have removed that comment from its original location and explained it more thoroughly in a new paragraph towards the end of the Discussion. The new paragraph reads as follows: “Our experiments, simulations, and theory focused on the thermodynamics of partitioning and do not directly explain the molecular details of how particles interact with condensates, nor how the particles diffuse into the condensates. However, related studies provide insights. Given that the hydrodynamic diameter of our particles significantly exceeds the mesh size of condensates (as measured by inert probes), particle diffusion into and within the condensates is unlikely to involve hopping between mesh cages.^{58–60} Instead, we speculate that on timescales longer than the characteristic relaxation times of the biopolymers within the condensates, these biopolymer chains diffuse and rearrange around the particles, potentially opening pathways for the particles to pass through. Indeed, intrinsically disordered proteins remain highly dynamic within condensates, with chain reconfiguration and exchange of interaction partners occurring on the sub-microsecond timescale⁶¹. An additional mechanism may be at play in some cases: the particles may compete for the same interactions that drive phase separation, which may locally open the condensate and allow the particle to become incorporated into the condensate. This would not likely occur for SA-RGG condensates with PS-PEG-biotin particles, because the particles bind to the proteins at domains not directly involved in driving phase separation, but it may well be relevant for the N

protein + RNA condensates with PS-PEG-polyA particles. As mentioned previously, a similar mechanism has been proposed to explain the translocation of nuclear transport receptors and cargo across the NPC, which is itself a dense protein phase composed of intrinsically disordered proteins.^{62,6364}”

2. Charge appears to be an important factor in ribosome partitioning, but it does not seem to be the only factor at play. This is evident since two of the model systems have the same net charge but exhibit different partitioning behaviors. A more detailed explanation for these differences is warranted.

We thank the reviewer for the suggestion. We have revised the paper in two ways based on this suggestion.

First, we have added Table S1 to highlight key features of the proteins we studied. From this table, the net charge per residue (NCPR) rather than net charge stands out as a clear distinguishing feature between the sequences that can explain our results. We revised the relevant portion of the results, and it now reads as follows: “These differences can be partially rationalized based on analysis of the three protein sequences (Table S1). The net charge per residue (NCPR) is positive for all three proteins and increases from RGG to N protein to (GRGNPYS)₂₅ (NCPR = 0.017, 0.053, and 0.109, respectively). Meanwhile, the ribosome surface has large areas with negative electrostatic potential^{32,33}.”

Protein	N	$\langle\lambda\rangle$	q	f+	f-	FCR	NCPR
RGG	176	0.571	3.0	0.136	0.119	0.255	0.017
N protein	433	0.586	23.0	0.139	0.085	0.224	0.053
GRGNPYS	229	0.637	25.0	0.122	0.013	0.135	0.109
SA-RGG	311	0.587	3	0.106	0.096	0.202	0.010

Table S1. Sequence statistics for four proteins used in this study. N is number of amino acids; λ is the average hydrophathy of the sequence based on the Urry hydrophathy scale (values range from 0 to 1, where 0 is the least hydrophobic); q is the net charge; f+ and f- are the fraction of positively and negatively charged residues in the sequence, respectively; FCR is the fraction of charged residues; and NCPR is the net charge per residue.

Second, we now state more clearly in the main text and Fig. 1 caption that the N protein (but not LAF-1 RGG or GRGNPYS) condensates were prepared with polyA RNA. This is expected to balance the charge of the N protein condensates and make ribosome partitioning less favorable. In light of our result in Fig. 4C, we decided to conduct a new experiment examining ribosome partitioning when N protein condensates are prepared without RNA, using 8 kDa PEG as a crowding agent. We have added new data comparing partitioning of ribosomes into N protein condensates prepared with PEG as crowding agent vs. RNA (SI Appendix Fig S7 and copied below). We show that ribosomes partition homogeneously throughout condensates formed from

N protein mixed with PEG. In contrast, condensates formed from N protein mixed with RNA shows inhomogeneous partitioning of ribosomes, with the ribosomes concentrating at the periphery of condensates. This indicates that competition for binding sites between ribosomes and polyA RNA also helps explain this question from the reviewer.

We have now added the following text to the Results section pertaining to Fig. 4C: “Motivated by this finding, we revisited the ribosome experiment, where we had previously observed inhomogeneous partitioning of ribosomes into N protein + 1x RNA condensates (Fig. 1B). We now tested ribosome partitioning into N protein condensates lacking RNA and using 8 kDa PEG as a crowding agent, resulting in robust and homogeneous partitioning of ribosomes in the condensates (SI Appendix, Fig. S7).”

Fig. S7. Partitioning of ribosomes into N protein condensates prepared with PEG vs. RNA. A) Ribosomes partition homogeneously into condensates prepared from N protein with 5% (w/v) 8 kDa PEG as a crowding agent. B) For condensates prepared with N protein + polyA RNA (the same conditions as in Fig. 1B; data copied here for the sake of comparison), ribosomes partition

inhomogeneously and are concentrated towards the periphery of the condensates. Fluorescence intensities were quantified as line profiles across individual condensates, which were normalized and averaged, for each image.

3. In Fig. 1C, the condensates display unusual "non-spherical shapes." What is the underlying cause of this behavior? Are these systems kinetically arrested?

Fig. 1C shows N protein condensates prepared with polyA RNA. Indeed, they are less uniformly circular than the LAF-1 RGG and the (GRGNPYS)₂₅ condensates. This is likely due to the material properties of N protein condensates. Specifically, N protein condensates are viscoelastic, whereas the other two condensates do not have a significant elastic component at the timescales measured. We have previously studied the material properties of RGG condensates in Schuster, Dignon et al., *PNAS* 2020 and (GRGNPYS)₂₅ condensates in Rekhi, Garcia, Barai et al, *Nat Chem* 2024. The rheology of N protein condensates has been characterized in a recent preprint from our lab, Favetta et al (<https://www.biorxiv.org/content/10.1101/2024.10.17.618867v1>), from which the following panel is taken, showing the frequency-dependent viscoelastic moduli of N protein + polyA RNA condensates:

We have now added a note to the first paragraph of the results: “The N protein condensates have an appreciable elastic rheological component and are therefore not uniformly circular.” We also cite our aforementioned papers.

On a related note, condensates prepared from N protein + polyA RNA are more viscoelastic compared to condensates prepared from N protein + PEG 8 kDa as a crowding agent. This data is presented in the aforementioned manuscript from Favetta et al. Consistent with this data, in Fig. S7 shown above, the N + PEG condensates are more spherical than the N + RNA condensates.

4. Most of the model systems in this study are positively charged, with cargos recruited via negatively charged interaction sites. While this surface chemistry seems straightforward in this context, it may be more complex for biological condensates. The authors should address how they would approach surface chemistry design in more complex systems. More generally, the limitations of the model systems used in this study should be discussed.

The reviewer is correct that the proteins used in this study have a net positive charge. However, as noted in Table S1, the proteins differ significantly in their net charge and NCPR. Furthermore, the results in Figs. 3 and 6 provide a compelling demonstration of distinct modes of partitioning. The partitioning of PS-PEG-biotin particles into SA-RGG condensates (Fig. 3) works based on the highly specific binding of biotin to streptavidin, as opposed to electrostatic interactions. This distinction is nicely highlighted in Fig. 6, which shows orthogonal partitioning of PS-PEG-biotin beads into SA-RGG condensates and targeting of PS-PEG-polyA20 beads into N protein condensates.

We agree with the reviewer that the PS-PEG-polyA beads would likely have poor specificity in the context of a complex cellular environment. In future studies, further fine-tuning of specificity, such as via sequence-specific bead functionalization, can be explored for applications such as drug delivery. We have now added the following sentences to the Discussion to note these limitations and point to future work: “Our study focused on particles functionalized with biotin or polyA, and examined their partitioning into three model condensates in vitro, but we expect that the approach and principles are generalizable. Besides those tested here, many other small molecule ligands and biopolymers could be used to target particles into a wide variety of condensates, even in more complex biological condensates. For instance, a logical extension is to functionalize particles with oligonucleotides or peptides of defined sequence to explore sequence-specific partitioning.”

5. For the biotin experiments (line 255), while the surface density remains constant for different bead sizes, the total number of available sites would vary due to the differences in bead size. Did the authors account for this in their experiments? If not, the trends reported in Fig. 3C may be more pronounced than observed.

We agree with the reviewer that if we kept the valence (number of biotins) the same while we increased particle size, we would expect partitioning to drop more steeply. However, in Fig. 3C, we chose to maintain constant biotin density (and hence increase the valence or number of biotins in proportion to the surface area) while varying particle size. Our rationale for designing the experiment this way is as follows:

First, it is consistent with how the simulations were conducted. In the simulations, in Fig. 2A, ϵ_{12} is held constant while bead size is increased. ϵ_{12} represents the bead-protein interaction energy, and the experimental analogue of holding ϵ_{12} constant is to hold the biotin density constant – meaning, the number of attached biotins will increase in proportion to the bead surface area.

Second, in Fig. 3C, since the larger particles have a greater valency and are still excluded, it is even more noteworthy of a finding. Fig. 3C is showing that larger particles with more interaction sites (proportional to their larger surface area) do not partition into condensates as well as smaller particles with fewer interaction sites. The derived theory provides a rationalization. Larger beads tend to be excluded because they displace attractive protein-protein interactions, which scales with volume (r^3). Meanwhile, if the biotin density is held constant, the number of interaction sites increases at a slower rate with the surface area of the particle (r^2).

We add the following text to explain this point after presenting the Fig. 3C results: “This agrees with the simulations at constant surface interaction strengths (Fig. 2A), where larger particles were excluded from the condensed phase despite having greater surface area and potential to interact with larger numbers of proteins. This is explained by the derived analytical theory, in which the attractive force driving inclusion of beads scales with bead surface area, while the excluding force arising from displacement of protein-protein interactions scales with bead volume.”

Reviewer #2 (Remarks to the Author):

Kelley et al study the partitioning of beads inside and around different types of condensates and show that both size and affinity of bead-protein interactions matter to determine the partitioning. This makes sense and it is somewhat obvious.

The paper is straightforward and with expected results, nothing particularly surprising or unexpected. The simulations with particles of different sizes and interaction strengths must have been done before in some form.

Despite that all this sounds negative, I think it deserves publication in Nat Comm as it is overall quite solid and well evidenced.

I don't find major flaws or poorly evidenced claims.

I can imagine that this paper will be a useful reference for future works on condensates.

We thank the reviewer for sharing the opinion that our work is well-evidenced, that it deserves publication in *Nature Communications*, and that it will be a useful reference for the field.

I have one important point that needs to be corrected before publication.

For microrheology to work, probe particles need to be NOT interacting with the polymers in solution. For this reason, I suggest the authors to modify their statements accordingly in the conclusions.

Specifically lines 446:

“One is the design of probes for microrheology measurements of condensates. It should be possible to ... condensate rheology”.

On this point, maybe the authors will find this review interesting for their current or future work <https://pubs.acs.org/doi/full/10.1021/jacsau.2c00055>

Thank you for sharing this excellent review article, which indeed is quite helpful.

The reviewer's concern is about our statement in the Discussion that “different functionalities could be used to test the effect of probe surface chemistry on condensate rheology.” Indeed, as the reviewer notes, valid microrheology measurements require that the probe experiences a continuum mechanical environment and does not alter that environment by interacting with the polymers in the material. Our intention was to argue for the value of future studies to examine how bead surface chemistry affects microrheology measurements of condensates, to identify optimal or suitable probes for condensate microrheology. This is in the spirit of Valentine ... Weitz et al, *Biophysical Journal*, 2004. It is also in the spirit of the following quote from the *Microrheology* textbook by Furst and Squires: “A number of ways have been used to

experimentally verify the validity of the continuum behavior in passive microrheology ... Another is to perform a series of experiments using different probe sizes, and a third is to use particles with different surface chemistries.”

Nevertheless, we understand that our statement may be confusing to readers given the overall focus of our manuscript on sticky particles. We have therefore deleted the statement from the manuscript.

Typos:

Line 134 “molecules larger than ???? would be ... “

We revised the sentence.

Line 460 "Our experiments demonstrated orthogonal, specific, and efficient partitioning of particles into condensates due to avidity" ? -> do they mean affinity?

We revised the sentence.

Reviewer #3 (Remarks to the Author):

The research of Kelley et al. investigates the partitioning of large particles into biomolecular condensates. Using polymer nanoparticles with tailored surface chemistries, the study demonstrates that strong particle-condensate interactions can drive the inclusion of even large (up to 1 μm) particles into condensates. The partitioning behavior can be tuned by altering factors like ionic strength, length of conjugated ligands, and density of sticky groups at nanoparticle surfaces. Orthogonal recruitment into immiscible condensates is also demonstrated. These findings offer valuable insights into the uptake of large client molecules by model condensates.

The authors present a carefully conducted investigation; experiments were performed with state-of-the-art techniques, and the quality of data and presentation of the results is very good. Their theoretical model satisfactorily recapitulates the experimental behavior.

The quality of experimental data allowed for the conclusive evaluation of how size and surface chemistry influence partitioning efficiency. While the study has undoubtful merits, it also has weaknesses that mitigate its relevance, as outlined below.

We are gratified that the reviewer noted the quality of our data and its presentation.

1) The manuscript fails to present the key determinants of phase separation for the representative molecules. What is the nature of scaffold-scaffold interactions? What are the main features of the respective condensate environments? Focusing on electrostatic interactions, the authors mention the overall net charges of the three selected molecules but disregard possible sequence or charge patterning contributions. It is unclear what differentiates one condensate from another and their representativeness. It appears that all condensates are driven by (partly) disordered polypeptide

chains, while the investigation does not include other types of phase-separating macromolecules such as proteins exhibiting multivalent modular domains. The common and divergent properties of the selected model systems should be better detailed. The presence of co-scaffold molecules (e.g. RNA) should be acknowledged from the very beginning.

We thank the reviewer for these suggestions. We have now added a new paragraph to the introduction to summarize the drivers of phase separation, the nature of scaffold-scaffold interactions, and the presence of co-scaffold molecules for the systems we studied. We have also cited therein several publications from our lab and others that explore these aspects in detail. The new paragraph reads as follows: “LAF-1 RGG is a low-complexity sequence rich in Gly, Arg, Asp, Asn, Ser, and Tyr. It has a near-neutral net charge and its phase separation is largely mediated by electrostatic interactions, hydrogen bonds, and cation- π and sp^2 - π interactions²². LAF-1 RGG is representative of intrinsically disordered regions common in biomolecular condensates²⁶. N protein is composed of a folded RNA-binding domain and a folded dimerization domain, interspersed among three intrinsically disordered domains⁵. N protein phase separates in association with RNA, and it is representative of RNA-binding proteins abundant in condensates. (GRGNPYS)₂₅ is a highly cationic, artificial intrinsically disordered protein comprising 25 repeats of the octapeptide GRGNPYS. In the presence of sufficient salt concentration to screen electrostatic repulsion, (GRGNPYS)₂₅ will phase separate, likely due to interactions of the Arg guanidinium group with Tyr and polar residues²⁶. (GRGNPYS)₂₅ is interesting because it has a large positive net charge yet does not require a polyanion to phase separate. Thus, these three proteins serve as distinct model systems for studying large-particle partitioning into condensates, and they exemplify three common types of proteins involved in condensate formation: intrinsically disordered domains, RNA-binding proteins, and de novo designed polypeptides.”

We added Supplemental Table 1 to compare descriptors of the proteins such as their net charge, hydrophobicity, and net charge per residue (NCPR). The table is copied above in the reply to Reviewer 1, question 2. As stated there, one notable difference between the sequences is the high NCPR for the (GRGNPYS)₂₅ sequence, which might partially explain the robust partitioning of ribosomes and PS-PEG-polyA20 beads into (GRGNPYS)₂₅ condensates.

We did not study multivalent modular domains, but that would be worthwhile to study in future work.

2) The theoretical framework proposed to explain the observed behavior provides somewhat limited insight into the chemistry and molecular rules underlying particle partitioning. The model predicts that stronger interactions will counteract particle exclusion of larger particles. Yet, the energetic contributions remain vague and undetailed; solvation effects are apparently not considered, or are they irrelevant? The recruitment of small molecules into certain condensates was described as an emergent property of the separated phase (Thody et al., Nat. Chem. 2024. <https://doi.org/10.1038/s41557-024-01630-w>). Is the recruitment of nanoparticles solely driven by the higher concentration of binding macromolecules within the dense phase (then simply a consequence of mass action), or are there additional, emergent properties at play?

The reviewer raises a good and very interesting point. The theoretical study involved in this work aims to offer general physical insights into the differing partitioning behaviors we observed, without being confined to specific molecular-level interactions.

With respect to the simulations, we justify and rationalize the simulation setup in the manuscript by adding the following text: “We note that since solvent is not explicitly considered in the simulations, the LJ interactions are purposed to implicitly account for the effect of solvent. One simplification that arises from this is that the protein-bead interactions are equivalent inside and outside the condensate, which may not reflect reality due to the difference in chemical environment inside the condensate^{6,8}. However, this should have minimal impact on the partitioning of beads in the simulation, since the beads form so few interactions with protein molecules in the dilute phase, and the interactions will be effectively modeling the interaction strength of protein-bead interactions inside the condensate. Stronger interactions in the simulation would thus represent beads with more interaction-prone surfaces, or more favorable interactions with particular proteins inside the condensate.”

So, yes, nanoparticle partitioning is driven by the higher concentration of binding molecules within the condensate phase. Particle partitioning may also be driven by emergent properties of the condensates, and the simulations indirectly account for that. Our experiments so far do not allow us to tease apart the role of emergent condensate properties, but we are very interested in studying that with future experiments.

3) The authors intend to demonstrate that larger particles can partition into condensates and that particles can be engineered for controlled partitioning:

“We hypothesized that for other clients, interactions with the scaffold protein can drive partitioning into condensates of clients > 5 nm” (line 110)

“These observations suggest that particles significantly larger than the dextrans may partition into condensates, depending on condensate-particle interaction strength.” (line 120)

“We therefore asked whether these interactions can be harnessed to engineer particles for controlled partitioning.” (line 268)

The use of large beads or particles to determine properties inside condensates is quite established (although in some cases they are embedded by microinjection) (e.g. Elbaum-Garfinkle et al., PNAS, 2015, <https://doi.org/10.1073/pnas.15048221>; JACS, 2019, Shayegan et al., <https://doi.org/10.1021/jacs.8b13349>; Avni et al., Nat Comm, 2022, <https://doi.org/10.1038/s41467-022-32143-0>; Pan et al., ACS Nano, 2022, <https://dx.doi.org/10.1021/acsnano.0c05486>). Moreover, there are examples in the literature that describe how surface functionalization with biomolecules can promote the inclusion of nanoparticles into biomolecular condensates driven by interactions (e.g. Barracchia et al., Bioconj. Chem. 2022, <https://doi.org/10.1021/acs.bioconjchem.2c00168>). Accordingly, the text should be modified to acknowledge that this is not a first-time discovery but rather one that advances previous investigations.

We thank the reviewer for the comment, and we have revised accordingly, adding these citations. Indeed, we view this paper not as the first to study particles in condensates, but rather our goal is to systematically study the roles of size and surface stickiness on particle partitioning into condensates. We added the following text to the Introduction: “Previous studies have used

nanoparticles that partition into condensates as probes of condensate material properties¹⁻⁴, and suitable surface functionalization of nanoparticles has been shown to promote particle recruitment into condensates⁵. However, a systematic investigation is required to test whether there is an upper limit to the size of clients that can partition into condensates, and to tease apart the biophysical principles that govern whether large clients partition into, are excluded from, or adsorb to the interface of condensates.”

4) The presented biomolecular condensates appear stable and unaffected by changes in experimental conditions. For example, high or low ionic strength conditions are not reported to perturb condensate stability. Did the authors test phase behavior in a range of salt concentrations? Supporting phase diagrams could be helpful.

We thank the reviewer for the suggestion. In brief, we chose conditions for our experiments where condensates will form. These conditions were based on prior studies from our lab and other labs that have mapped the phase behavior of the proteins of interest. We have added the following sentence to the Results relating to Fig. 4D: “Low salt concentration promotes phase separation of LAF-1 RGG, but high salt concentration promotes phase separation of (GRGNPYS)₂₅; we selected suitable buffer conditions for phase separation based on prior characterization of these proteins^{17,26}. Despite the different phase behavior of these two proteins, in both cases, lower salt concentration favored PS-PEG-polyA20 particle partitioning.” The references cited include phase diagrams showing the salt-dependent phase behavior.

Here are additional details: The phase behavior of (GRGNPYS)₂₅ in a range of salt conditions has previously been explored (copied below from Extended Fig. 1 in Rekhi, Garcia, Barai et al., *Nat Chem*, 2024; <https://www.nature.com/articles/s41557-024-01489-x#Sec29>). The propensity to phase separate increases monotonically with NaCl concentration.

The coexistence curve of LAF-1 (full length and RGG domain) in a range of salt conditions has also been previously explored (copied below from Fig. S7 in Elbaum-Garfinkle et al, *PNAS*, 2015). The critical protein concentration for RGG phase separation decreases with salt concentration.

Figure Redacted

Moreover, do the authors have any evidence about the impact of nanoparticle exposure on condensate size and stability?

High concentrations of nanoparticles may affect condensate stability, but low concentrations do not. To illustrate this point, we have added a new supplemental figure, Fig. S3, copied below. Throughout our experiments, we picked bead concentrations that were fairly low (0.02 vol% or below) and did not affect condensate size, stability, or morphology.

Fig. S3. Effect of bead concentration and surface chemistry on condensate morphology and stability. LAF-1 RGG condensates were mixed with different concentrations of PS and PS-PEG beads (500 nm) and imaged after 10 minutes and after 24 hours. RGG condensates appear to aggregate when exposed to high (0.1 vol%) concentration of the sticky PS beads, whereas high concentration of PS-PEG beads do not appear to affect condensate morphology. Lower bead concentrations (0.02 vol%) did not appear to alter condensate morphology or stability.

5) The morphological characterization of the particles used in the experiments needs to be implemented. Hydrodynamic diameter data must be presented with a more appropriate number of significant figures. It is unclear whether the error reported for these measurements is the standard deviation over a bunch of repeated measurements, or if it is the standard deviation of particle size distribution in each sample. In the latter case, certain values appear unrealistic (200

nm PS-PEG-azide 199 ± 0.06 nm; 100 nm PS-PEG-30% biotin 140 ± 0.56 nm; 200 nm PS-PEG-20% biotin 211 ± 0.09 nm). It would be useful to report in Table S1 the polydispersion index values for these measurements to represent the sample size distribution width more clearly. It would also be helpful to show hydrodynamic values under the ionic strength conditions in which the particle samples have been tested (50, 150 mM, and sometimes 900 mM) rather than in pure water.

Similar considerations apply to Z-potential measurements. Values should be reported with an appropriate number of significant figures. The same particles are often used in experiments with biomolecular condensates under very different ionic strength conditions (50, 150 mM, sometimes 900 mM). Since Z-potential can be affected by ionic strength, it would be appropriate to show Z potential values under the ionic strength conditions in which the particle samples have been tested and not just in 0.1X PBS buffer.

We thank the reviewer for this feedback. In response, we have updated the supplemental tables by improving our presentation of the data, and by conducting additional size and zeta-potential measurements.

First, we revised Table S2 to show the polydispersity index (PDI), in addition to the standard deviation values. Size and standard deviation of size are based on $n = 2$ independent measurements, as is the PDI. We also measured the size of beads in different salt concentrations. Size measurements were relatively consistent over different salt conditions.

Measurements in water

Particle	Size (nm)	Standard Deviation (SD) (nm)	Polydispersity index (PDI)
200 nm PS	220	3.7	0.027
200 nm PS-PEG-azide	200	0.063	0.037
500 nm PS	570	25	0.11
500 nm PS-PEG-azide	620	12	0.11
1 μ m PS	1200	140	0.10
1 μ m PS-PEG-azide	1100	52	0.061
500 nm PS-PEG-polyA20	620	52	0.097
500 nm PS-PEG-polyA40	540	6.0	0.024
500 nm PS-PEG-10% polyA20	600	13	0.20
500 nm PS-PEG-75% polyA20	620	9.1	0.12
500 nm PS-PEG-polyA20/T20	640	5.6	0.13
1 μ m PS-PEG-polyA20/T20	1100	38	0.063
100 nm PS-PEG-biotin	140	0.56	0.011
200 nm PS-PEG-biotin	210	0.093	0.025
500 nm PS-PEG-10% biotin	530	17	0.037
500 nm PS-PEG-100% biotin	650	31	0.10
200 nm PS-PEG-polyA20	230	2.5	0.029

Measurements in 50 mM NaCl, 20 mM Tris, pH 7.5

Particle	Size (nm)	SD (nm)	PDI
500 nm PS-PEG-azide	540	21	0.10
500 nm PS-PEG-polyA20	620	59	0.17

Measurements in 150 mM NaCl, 20 mM Tris, pH 7.5

Particle	Size (nm)	SD (nm)	PDI
500 nm PS-PEG-azide	570	28	0.064
500 nm PS-PEG-polyA20	640	110	0.093

Measurements in 900 mM NaCl, 20 mM Tris, pH 7.5

Particle	Size (nm)	SD (nm)	PDI
500 nm PS-PEG-azide	570	6.6	0.11
500 nm PS-PEG-polyA20	580	15	0.13

Table S2. Size measurements of beads used in this study. Particle size was measured by dynamic light scattering. Particle size stated in the leftmost column is the nominal size reported by the manufacturer. Size and standard deviation (SD) of size are based on $n = 2$ independent measurements, as is the polydispersity index (PDI). For reference, $PDI < 0.1$ is indicative of a monodisperse sample, $PDI 0.1 - 0.4$ is indicative of a moderately polydisperse sample, and $PDI > 0.4$ is indicative of a broad polydisperse sample.

Second, we also measured the zeta potential of beads in different buffer conditions in SI Appendix Table 3 (below). We measured reduced magnitude of zeta potential values at 50 mM NaCl and above.

Measurements in 0.1x PBS

Particle	Charge (mV)	SD (mV)
200 nm PS	-51	5.3
200 nm PS-PEG-azide	0.11	0.40
500 nm PS	-66	1.7
500 nm PS-PEG-azide	-0.51	0.66
500 nm PS-PEG-10% polyA20	-4.7	5.8

500 nm PS-PEG-75% polyA20	-29	1.7
500 nm PS-PEG-100% polyA20	-34	4.7
500 nm PS-PEG-polyA5	-16	5.3
500 nm PS-PEG-polyA40	-37	5.8
500 nm PS-PEG-polyA20/T20	-28	0.97

Measurements in 50 mM NaCl, 20 mM Tris, pH 7.5

Particle	Charge (mV)	SD (mV)
500 nm PS-PEG-azide	-0.97	1.6
500 nm PS-PEG-100% polyA20	-16	4.6

Measurements in 150 mM NaCl, 20 mM Tris, pH 7.5

Particle	Charge (mV)	SD (mV)
500 nm PS-PEG-azide	-1.8	0.14
500 nm PS-PEG-100% polyA20	-13	0.41

Measurements in 900 mM NaCl, 20 mM Tris, pH 7.5

Particle	Charge (mV)	SD (mV)
500 nm PS-PEG-azide	3.6	0.64
500 nm PS-PEG-100% polyA20	-12	3.9

Table S3. Zeta potential measurements of beads used in this study. Zeta potential was measured by laser doppler electrophoresis. Numbers represent average of $n = 2$ independent measurements. SD is the standard deviation. Zeta potential of PS-PEG-azide and PS-PEG-100%polyA20 beads were measured in various buffer conditions used in this study.

Minor points:

6) The terminology of the following sentence is specialized: “We reason that this hypothesis will be valid if the condensate is terminally viscous, so that the biopolymer network can flow around the particles at time scales longer than the reptation relaxation time”. Please add an explanation.

We thank the reviewer for the comment. We decided to delete this sentence from the Introduction, and instead we expounded upon this concept in more detail in the second-to-last paragraph of the Discussion. Please see our reply to Reviewer 1, Question 1.

7) “Absence of RNA led to more robust partitioning of PS-PEG-polyA beads into condensates, whereas 2x RNA concentration resulted in >98% of PS-PEG-polyA beads being excluded from the condensates (Fig. 4C). This result demonstrates that “free” nucleic acids can compete with particle-conjugated nucleic acids for binding to N protein, and hints at a possible mechanism by which condensates exclude unwanted clients in cells.” These findings appear obvious, therefore this part is not very informative.

We have conducted additional experiments on ribosome partitioning into N protein condensates prepared with RNA vs. with PEG as a crowding agent. We added new text to this paragraph of the Results and included a new Fig. S7. Please see our reply to Reviewer 1, question 2 for details. With these edits, the paragraph is now more informative.